

# Cranial ontogenetic variation in early saurischians and the role of heterochrony in the diversification of predatory dinosaurs

Christian Foth[1,2,3], Brandon P. Hedrick[4,5] and Martin D. Ezcurra[2,6,7]

[1] SNSB, Bayerische Staatssammlung für Paläontologie und Geologie, München, Germany
[2] Department of Earth and Environmental Sciences, Ludwig-Maximilians-Universität, München, Germany
[3] Department of Geosciences, University of Fribourg/Freiburg, Fribourg, Switzerland
[4] Department of Earth and Environmental Science, University of Pennsylvania, Philadelphia, PA, United States
[5] Department of Biology, University of Massachusetts, Amherst, MA, United States
[6] CONICET, Sección Paleontología de Vertebrados, Museo Argentino de Ciencias Naturales, Buenos Aires, Argentina
[7] School of Geography, Earth and Environmental Sciences, University of Birmingham, Birmingham, United Kingdom

## ABSTRACT

Non-avian saurischian skulls underwent at least 165 million years of evolution and shapes varied from elongated skulls, such as in the theropod *Coelophysis*, to short and box-shaped skulls, such as in the sauropod *Camarasaurus*. A number of factors have long been considered to drive skull shape, including phylogeny, dietary preferences and functional constraints. However, heterochrony is increasingly being recognized as an important factor in dinosaur evolution. In order to quantitatively analyse the impact of heterochrony on saurischian skull shape, we analysed five ontogenetic trajectories using two-dimensional geometric morphometrics in a phylogenetic framework. This allowed for the comparative investigation of main ontogenetic shape changes and the evaluation of how heterochrony affected skull shape through both ontogenetic and phylogenetic trajectories. Using principal component analyses and multivariate regressions, it was possible to quantify different ontogenetic trajectories and evaluate them for evidence of heterochronic events allowing testing of previous hypotheses on cranial heterochrony in saurischians. We found that the skull shape of the hypothetical ancestor of Saurischia likely led to basal Sauropodomorpha through paedomorphosis, and to basal Theropoda mainly through peramorphosis. Paedomorphosis then led from Orionides to Avetheropoda, indicating that the paedomorphic trend found by previous authors in advanced coelurosaurs may extend back into the early evolution of Avetheropoda. Not only are changes in saurischian skull shape complex due to the large number of factors that affected it, but heterochrony itself is complex, with a number of possible reversals throughout non-avian saurischian evolution. In general, the sampling of complete ontogenetic trajectories including early juveniles is considerably lower than the sampling of single adult or subadult individuals, which is a major impediment to the study of heterochrony on non-avian dinosaurs. Thus, the current work represents an exploratory analysis. To better understand the cranial ontogeny and the impact of heterochrony on skull evolution in saurischians, the data set that we present here must be expanded and complemented with further sampling from future fossil discoveries, especially of juvenile individuals.

Corresponding author
Christian Foth, christian.foth@gmx.net

# INTRODUCTION

In an evolutionary context, heterochrony describes phenotypic changes due to shifts in the timing or rate of developmental processes in an organism relative to its ancestor, and can lead to significant evolutionary changes in body plans within relatively short periods of time (*Gould, 1977*; *Alberch et al., 1979*; *McNamara, 1982*; *Reilly, Wiley & Meinhardt, 1997*; *Klingenberg, 1998*; *McNamara & McKinney, 2005*). Two major types of heterochronic processes are discerned: paedomorphosis and peramorphosis. Paedomorphosis occurs when the later ontogenetic stages of an organism retain characteristics from earlier ontogenetic stages of its ancestor due to a truncation of the growth period (progenesis), decrease of the growth rate (neoteny) or a delayed onset of developmental processes (postdisplacement). In contrast, a peramorphic organism is ontogenetically more developed than the later ontogenetic stages of its ancestor due to the extension of growth period (hypermorphosis), the increase of the growth rate (acceleration) or the earlier onset of developmental processes (predisplacement) (see *Gould, 1977*; *Alberch et al., 1979*; *Klingenberg, 1998*). In practice, evidence for heterochronic events in evolution can be detected by comparing the ontogenetic trajectories of different taxa under the consideration of their phylogenetic interrelationships (*Alberch et al., 1979*; *Fink, 1982*). Thus, the concept of heterochrony connects two main fields of biological sciences: developmental and evolutionary biology (*Gould, 1977*; *Raff, 1996*). When studying heterochrony, ontogenetic trajectories are characterized by three separate vectors (size, shape, and ontogenetic age), which allows for quantification of heterochronic processes with slope, length and position within a Euclidean space (*Alberch et al., 1979*). In this context, geometric morphometrics is a useful method for characterizing shape and size vectors to investigate heterochrony in organisms within a multivariate framework (*Mitteroecker, Gunz & Bookstein, 2005*).

Documentation of heterochrony in the vertebrate fossil record is limited. Preserved fossil ontogenetic series covering the whole postnatal development of fossil species are rare due to the fact that early juvenile specimens are often either lacking or incomplete. Furthermore, exact ages of single ontogenetic stages are often not available, resulting in the temporal component often being replaced by size, which is not an ideal variable for age (*Klingenberg, 1998*; *Gould, 2000*). Nevertheless, the role of heterochrony has been recognized and discussed for the evolution of multiple fossil lineages that do preserve ontogenetic series (*Balanoff & Rowe, 2007*; *Gerber, Neige & Eble, 2007*; *Schoch, 2009*; *Schoch, 2010*; *Schoch, 2014*; *Bhullar, 2012*; *Forasiepi & Sánchez-Villagra, 2014*; *Ezcurra & Butler, 2015*), including non-avian dinosaurs (e.g., *Long & McNamara, 1997*; *Erickson et al., 2004*; *Guenther, 2009*; *Bhullar et al., 2012*; *Canale et al., 2014*). For example, *Long & McNamara (1997)*, *Erickson et al. (2004)* and *Canale et al. (2014)* hypothesized that the evolution of large body size in carcharodontosaurids and tyrannosaurids from medium-sized ancestors was the result of peramorphosis.

There has recently been an increasing interest in shape diversity in non-avian dinosaurs, in which geometric morphometric methods have been applied on a regular basis (e.g., *Bonnan, 2004*; *Chinnery, 2004*; *Campione & Evans, 2011*; *Hedrick & Dodson, 2013*; *Lautenschlager, 2014*; *Schwarz-Wings & Böhm, 2014*; *Maiorini et al., 2015*). Skull shape diversity in saurischian dinosaurs has been studied in particular detail (e.g., *Henderson, 2002*; *Young & Larvan, 2010*; *Rauhut et al., 2011*; *Brusatte et al., 2012*; *Bhullar et al., 2012*; *Foth & Rauhut, 2013a*; *Foth & Rauhut, 2013b*), but usually in relation to functional constraints, dietary preferences, phylogenetic interrelationships, and macroevolutionary patterns. For example, these studies have shown that skull shape in sauropodomorphs and theropods is phylogenetically constrained (*Young & Larvan, 2010*; *Brusatte et al., 2012*; *Foth & Rauhut, 2013a*) and that the shape of the orbit in theropods is functionally constrained (*Henderson, 2002*; *Foth & Rauhut, 2013a*). Thus, geometric morphometrics is a powerful method to quantify both intraspecific (e.g., ontogeny, sexual dimorphism, polymorphism) and interspecific (e.g., systematics, macroevolution) shape variation on the basis of homologous landmarks or outlines, which capture more information about shape than traditional morphometric measurements (*Corti, 1993*; *Rohlf & Marcus, 1993*; *Adams, Rohlf & Slice, 2004*; *Adams, Rohlf & Slice, 2013*; *Slice, 2007*; *Mitteroecker & Gunz, 2009*; *Zelditch, Swiderski & Sheets, 2012*). As a result, geometric morphometrics has also been successfully applied to the study of heterochrony among various tetrapod groups, in which the univariate mathematical approach of *Alberch et al. (1979)* was adapted to a multivariate framework (e.g., *Berge & Penin, 2004*; *Mitteroecker et al., 2004*; *Mitteroecker, Gunz & Bookstein, 2005*; *Lieberman et al., 2007*; *Drake, 2011*; *Piras et al., 2011*; *Bhullar et al., 2012*). However, only *Bhullar et al. (2012)* have examined cranial shape diversity of theropod dinosaurs using multivariate methods in the context of heterochrony. This pioneering study demonstrated that recent birds have highly paedomorphic skulls compared to non-avian theropods and Mesozoic birds (e.g., *Archaeopteryx* and Enantiornithes), which evolved in a multistep transformation within the clade Eumaniraptora. Furthermore, *Bhullar et al. (2012)* found evidence for independent peramorphic trends in the skull shape of large-bodied tyrannosaurids, dromaeosaurids and troodontids and proposed a similar trend for allosaurids. Finally, *Bhullar et al. (2012)* hypothesized a possible paedomorphosis for *Eoraptor* and basal sauropodomorphs.

The aim of the current study is to investigate the cranial shape diversity of saurischian dinosaurs by comparing the ontogenetic trajectories of different taxa from both qualitative and quantitative data, using two-dimensional geometric morphometrics (2D GM). This study expands on the work of *Bhullar et al. (2012)* who focused primarily on trends within Maniraptora, derived non-avian theropods and basal avian theropods. We have built upon their study by including an improved sample of basal saurischians and theropods (including a number of different ontogenetic series), which should be more sensitive for testing of the heterochronic changes for allosaurids and basal sauropodomorphs proposed, but not verified statistically, by *Bhullar et al. (2012)*. The phylogenetic relationships of the ontogenetic series sampled in this study are integrated into an ancestor-descendant framework to look for further potential heterochronic processes in the cranial evolution of saurischians. However, due to the limited number of ontogenetic series known for

sauropodomorphs, the current study focuses primarily on the early evolution of theropods. Nevertheless, due to the limited number of ontogenetic series currently available in our taxonomic sample, this work must be viewed as an exploratory study, which will need to be expanded and complemented with further sampling from future fossil discoveries.

## MATERIALS AND METHODS

### Taxon sampling

We sampled the crania of 35 saurischian dinosaur taxa (10 sauropodomorphs and 25 non-pennaraptoran theropods, see Table S3) on the basis of published reconstructions of adult (or advanced subadult) individuals in lateral view (with exception of the reconstructions of the basal tyrannosauroid *Dilong* [IVPP V14243] and the basal alvarezsauroid *Haplocheirus* [IVPP V15988], which were based on our personal observations). The data set shows an overlap of 15 terminal taxa with that of *Bhullar et al. (2012)* and builds on that study with an addition of 20 new taxa. Theropods with large nasal crests (e.g., *Ceratosaurus*, *Dilophosaurus*, *Guanlong*) were excluded from the primary data set as they were found to have a strong impact on the ancestral shape reconstruction (see below) of Averostra, Avetheropoda, Coelurosauria and Tyrannosauroidea (see Fig. S5 and Table S6). Although cranial crests are a common structure among theropod dinosaurs (*Molnar, 2005*), reconstruction of moderately to strongly crested hypothetical ancestors within this study would necessarily be artificial due to the lack of intermediate crested forms and relatively small sample size of the available data set. Only *Monolophosaurus* was included in the main data set because it possesses a rather moderately sized and simple nasal crest. '*Syntarsus' kayentakatae*, which is often reconstructed with a pair of prominent nasal crests (*Rowe, 1989*; *Tykoski, 1998*), was analysed in this study without crests since this structure is probably artificial due to post-mortem displacement of the nasals (*Ezcurra & Novas, 2005*; *Ezcurra & Novas, 2007*). As cranial crests usually represent external visual signal structures (*Sampson, 1999*; *Padian & Horner, 2011*; *Hone, Naish & Cuthill, 2012*), their evolutionary development most likely represents either an evolutionary novelty or was sourced from regional peramorphic processes if the primordia were already present in the ancestor (see discussion on the evolution of horns and frills in Ceratopsia by *Long & McNamara (1997)*). However, we generated a second data set that includes crested taxa for comparison with the main data set (see below).

In our sample, five taxa preserve early ontogenetic stages allowing the capture of both juvenile and adult skull shapes, which were used to reconstruct five simplified ontogenetic series, containing two stages (i.e., an early juvenile and adult stage). This sample includes the basal sauropodomorph *Massospondylus*, the basal theropod *Coelophysis*, the megalosaurid *Dubreuillosaurus*, the allosauroid *Allosaurus*, and the tyrannosaurid *Tarbosaurus* (see Table S4). Two of the ontogenetic series sampled (*Coelophysis* and *Tyrannosaurus/Tarbosaurus*) overlap with the data set from *Bhullar et al. (2012)*, but we expand on the previous study by including three more basal trajectories in order to concentrate on a different part of the theropod tree. As the fossil record of juvenile dinosaur specimens with complete skull material is rare, the number of ontogenetic series is limited. To improve

sampling, previous studies have included reconstructions from multiple partial juvenile skulls or juveniles from closely related taxa (e.g., *Bhullar et al., 2012*). We implemented this approach in two cases: the reconstruction of the juvenile *Coelophysis* sample was based on three incomplete, somewhat taphonomically deformed individuals (MCZ 4326; GR 392; CM 31375); and the holotype of *Sciurumimus* (BMMS BK 11) was used as the juvenile representative of the megalosaurid *Dubreuillosaurus* based on the phylogenetic analyses of *Rauhut et al. (2012)*. In contrast to *Bhullar et al. (2012)*, we did not include the ontogenetic series of *Byronosaurus*, Therizinosauridae (represented by a therizinosaurid embryo and the skull of *Erlikosaurus*) and *Compsognathus* (with the juvenile specimen represented by *Scipionyx*) in the data set because the postorbital region of the juvenile skulls of the former two taxa is crushed or incomplete (*Bever & Norell, 2009*; *Kundrát et al., 2008*), and the taxonomic referral of *Scipionyx* to the clade Compsognathidae (see *Dal Sasso & Maganuco, 2011*) is uncertain and maybe an artefact of coding juvenile character states (see *Rauhut et al., 2012*).

## Two-dimensional geometric morphometrics (2D GM)

We used 20 landmarks (LMs) and 51 semi-landmarks (semi-LMs) on our sample in order to accurately capture skull shape. The landmarks were collected using the software tpsDig2 (*Rohlf, 2005*) and were classified as either type 1 (points where two bone sutures meet) or type 2 (points of maximum curvature and extremities) (*Bookstein, 1991*) (see Fig. S1 and Table S1 for full description). Type 3 landmarks (points constructed between two homologous landmarks, which mainly define the shape of the skull or skull openings rather than the position of exact homologous points) were not used in our study. Semi-landmarks were used to capture the shape of skull openings and the skull outline by defining a number of points that are placed equidistantly along respective curves (*Bookstein, 1991*; *Bookstein et al., 1999*). The percent error for digitizing each landmark and semi-landmark was estimated for the skull reconstruction of the juvenile *Coelophysis* (with $n = 10$ replications) using the method described by *Singleton (2002)*. Landmark and semi-landmark error varies between 0.117 percent (LM 51—most posterior point of the descending process of the maxilla contacting the nasal and/or the lacrimal) and 0.738% (LM 3—contact between the maxilla and jugal along the ventral margin of the skull) with a mean of 0.283%. The error has no significant effect on the shape analyses (see Table S2).

The shape coordinates were then imported into the software package MorphoJ 1.05d (*Klingenberg, 2011*) and superimposed using generalized Procrustes analysis (GPA). GPA rotates, translates and resizes landmark coordinates of all specimens accounting for all non-shape related differences between landmark configurations, leaving only shape information (*Gower, 1975*; *Rohlf & Slice, 1990*). Although semi-landmarks have fewer degrees of freedom than regular landmarks (and thus contain less shape information) (*Bookstein, 1991*), we treated landmarks and semi-landmarks as equivalent for GPA (*Zelditch, Swiderski & Sheets, 2012*) and did not slide the semi-landmarks. The sliding process created considerable artificial deformation on the Procrustes-fitted shape in some taxa (see Fig. S2). However, due to the equivalent weighting of landmarks and semi-landmarks, it should be kept in mind that the shape information captured by the

semi-landmarks strongly influences the results (*Zelditch, Swiderski & Sheets, 2012*; see below). In order to estimate the influence of the semi-landmarks on the shape data, all analyses described below were also applied to an additional data set that included only landmark data (see Supplemental Information).

The generated Procrustes coordinates were used to compare juvenile and adult skull shapes to each other in each ontogenetic series to find ontogenetic patterns between and within taxa. Furthermore, the Procrustes coordinates of all taxa (including juvenile specimens) were subjected to an exploratory principal components analysis (PCA) using the covariance matrix generated from Procrustes coordinates. PCA simplifies descriptions of variation among individuals by creating new sets of variables that are linear combinations of the original set such that the new sets are independent from one another and have zero covariance. The principal components (PCs) describe successively smaller amounts of total variance of the sample. This allows for a larger proportion of the variance to be described using a smaller number of variables than the original data would have allowed (*Zelditch, Swiderski & Sheets, 2012*). A multivariate regression of the Procrustes coordinates against log-transformed centroid sizes (=square root of the sum of the squared distances of each landmark to the centroid of the landmark configuration, *Zelditch, Swiderski & Sheets, 2012*) was performed to test if the skull shape variation is correlated with size and contains allometric information (*Drake & Klingenberg, 2008*).

## Quantification of ontogenetic trajectories

The different ontogenetic trajectories generated in the PCA and regression analyses were compared to each other by calculating pairwise two-dimensional angles between different trajectories based on the PC values of the first three axes, which are the significant principal components (significance calculated using the broken stick method, see *Jackson, 1993*). Each of the two-stage ontogenetic trajectories was described as a phenotypic change vector, $\Delta \vec{y}_i = \vec{y}_{ij} - \vec{y}_{ik}$, with two shape traits (PC 1 vs. PC 2 and PC 1 vs. PC 3), where $i$ is a specific ontogeny between two fixed stages, juvenile ($j$) and adult ($k$) (*Collyer & Adams, 2007*). The difference in direction (angle) between the ontogenetic phenotypic change vectors $\Delta \vec{y}_a, \Delta \vec{y}_b$ was calculated using the dot product $\cos^{-1}(\Delta \vec{y}_a, \Delta \vec{y}_b) = \frac{\Delta \vec{y}_a \cdot \Delta \vec{y}_b}{|\Delta \vec{y}_a||\Delta \vec{y}_b|}$. PC values were employed to calculate the length of each ontogenetic trajectory. Lengths and angles were used to characterise the differences between the ontogenetic trajectories in relation to shape variation.

## Phylogenetic framework for heterochronic analyses

In an evolutionary context, heterochrony is defined as the change in the timing or rate of developmental processes in ancestor-descendant relationships (*Alberch et al., 1979*; *Fink, 1982*; *Klingenberg, 1998*), and thus a direct comparison of ontogenetic trajectories from different species (as terminal taxa) can be problematic because it is hard to determine which trajectory would represent the ancestral and the descendant form, respectively (see *Fink, 1982*). This is exacerbated when the supposed ancestral (terminal) species possesses an unknown, long evolutionary history resulting from a ghost lineage. This problem can be partially solved using a phylogenetic approach (see *Alberch et al., 1979*;

*Fink, 1982*; *Balanoff & Rowe, 2007*; *Bhullar, 2012*; *Fritsch, Bininda-Emonds & Richter, 2013*; *Mallon, Ryan & Campbell, 2015*), in which the ancestor of two sister (terminal) taxa is represented by the hypothetical last common ancestor (*Hennig, 1966*). Therefore, on the basis of the phylogenetic distribution of the five ontogenetic series sampled we calculated hypothetical ancestral ontogenetic trajectories for Saurischia, Neotheropoda, Orionides and Avetheropoda using ancestral shape reconstructions as follows (see Figs. S3 and S4). An informal supertree (sensu *Butler & Goswami, 2008*) including all taxa with adult individuals was created based on recent phylogenetic analyses (see Figs. S3 and S4): basal Sauropodomorpha (*Cabreira et al., 2011*), Coelophysoidea (*Ezcurra & Novas, 2007*), Ceratosauria (*Pol & Rauhut, 2012*), Tetanurae (*Carrano, Benson & Sampson, 2012*), and Coelurosauria (*Turner, Makovicky & Norell, 2012*; *Loewen et al., 2013*). The phylogenetic position of *Eoraptor* follows *Martínez et al. (2011)* and *Martínez, Apaldetti & Abelin (2013)*. The position of *Adeopapposaurus* as sister taxon of *Massospondylus* follows *Martínez (2009)*. The position of *Herrerasaurus* and *Tawa* at the base of Theropoda is based on *Sues et al. (2011)*. *Zupaysaurus* was placed outside Coelophysoidea as one of the successive sister taxa of Averostra (*Smith et al., 2007*; *Sues et al., 2011*; *Ezcurra, 2012*). The supertree was time-calibrated using the stratigraphic age of each taxon (as mean of time interval) (see Tables S3 and S5). The assignment of branch lengths was performed in R (*R Development Core Team, 2011*) using the APE package (version 2.7-2; *Paradis, Claude & Strimmer, 2004*) and a protocol written by Graeme Lloyd (see http://www.graemetlloyd.com/methdpf.html) for adjusting zero branch lengths by sharing out the time equally between branches (see *Brusatte et al., 2008*; *Brusatte, 2011*), and adding an arbitrary length of 1 million years to the root. The time-calibrated supertree was imported into the software package Mesquite 2.72 (*Maddison & Maddison, 2009*). Subsequently, Procrustes coordinates and centroid sizes of the adult representatives of the taxa were mapped onto the supertree as continuous characters using square change parsimony. This algorithm performs an ancestral state reconstruction by collating the sum of squared changes of continuous characters along all branches of a tree and estimates the most parsimonious ancestral states by minimizing the total sum of squared changes across the tree (*Maddison, 1991*). In the next step we tested if the continuous data contains a phylogenetic signal. We performed a permutation test in MorphoJ in which the topology was held constant and both the Procrustes-fitted shape data and the centroid size for each taxon were randomly permuted for all the terminals across the tree 10,000 times (*Laurin, 2004*; *Klingenberg & Gidaszewski, 2010*). The data are considered to contain a statistically significant phylogenetic signal if the squared length of the original supertree occurs in at least 95% of the randomly generated trees. Additionally, we quantified phylogenetic signal in our data using a multivariate form of the $K$ statistic with 10,000 replications (*Blomberg, Garland & AR, 2003*; *Paradis, 2012*; *Adams, 2014*) in R using the package geomorph (*Adams & Otárola-Castillo, 2013*). This test estimates the strength of a phylogenetic signal in a data set in relation to a simulated Brownian motion model, which is expressed as $K$ and $p$ values.

To obtain ancestral ontogenetic trajectories, the protocol described above was repeated in a new nexus file containing the Procrustes-fitted shapes and centroid sizes of the juvenile

specimens. As the juvenile data set is only represented by five taxa, the original supertree was pruned such that only these taxa remained, retaining the original time-calibration. Finally, the ancestral Procrustes-fitted shapes and centroid sizes of both juvenile and adult Saurischia, Neotheropoda, Orionides and Avetheropoda were exported and combined with the respective data from the ontogenetic trajectories of the terminal taxa. The ancestral Procrustes-fitted shape of Averostra was not considered because no ceratosaur juveniles have been published in detail so far (see *Madsen & Welles, 2000*). The new data set including the five terminal and four ancestral ontogenetic trajectories was loaded again into MorphoJ.

## Regression analyses of ontogenetic trajectories

A multivariate, pooled within-group regression of shape against log-transformed centroid size including terminal taxa and hypothetical ancestors (see above) was performed (*Piras et al., 2011*; *Bhullar et al., 2012*; *Zelditch, Swiderski & Sheets, 2012*), in which the Procrustes coordinates were transformed into a regression score (see *Drake & Klingenberg, 2008*). In contrast to many previous studies of heterochrony using geometric morphometrics, which compare only the ontogenetic trajectories of terminal taxa, our approach allows the determination of possible heterochronic patterns between ancestors and descendants. The different ontogenetic trajectories were compared regarding slope, length, angles and range of shape variation spanned by the predicted regression score. The angles between ontogenetic trajectories were calculated based on Procrustes distances and centroid sizes (see above).

As mentioned above, studies of heterochrony require size, shape and ontogenetic age as independent vectors (*Klingenberg, 1998*). Due to missing data on the individual age of the specimens, ontogenetic age could not be taken into account. As a consequence, the regression analysis explores allometry and not heterochrony (*Klingenberg & Spence, 1993*; *Klingenberg, 1998*; *Gould, 2000*). While some heterochronic processes can result from allometric changes (e.g., acceleration and neoteny), allometric studies allow only conclusions regarding paedomorphosis and peramorphosis (*Klingenberg & Spence, 1993*; *Klingenberg, 1998*), which are expressed by the shape vector (i.e., regression score). Peramorphosis can be inferred if the adult individual of the descendant trajectory falls along higher regression scores than the respective ancestral one, whereas paedomorphosis can be inferred based along lower scores. To verify the results of such regression analyses we repeated the analysis using Euclidean distance, which is equivalent to Procrustes distance (see *Singleton, 2002*; *Tallman et al., 2013*) as a separate shape vector measuring differences in shape. The Euclidean distance matrix was calculated in PAST 3.05 (*Hammer, Harper & Ryan, 2001*) on the basis of the Procrustes coordinates of terminal taxa and hypothetical ancestors (see above), which were exported from MorphoJ. For regression analysis, the juvenile specimen of *Massospondylus*, which represents the sample with the smallest centroid size, was set to zero for aligning the distance values of the remaining taxa (Fig. 4).

To test if the shape changes, and as a result the presence of heterochrony, of an ancestor-descendant relationship are statistically meaningful, we calculated the confidence interval (CI) of the differences between regression scores and Euclidean distances of terminal and ancestral taxa ($n = 68$) and compared them with the differences of ancestral and descendant

regression scores from the sub-sample containing the ontogenetic trajectories. Changes were considered significant if the differences between regression scores were at least 1.5 times higher than the CI value (see *Cumming, Fidler & Vaux, 2007*).

For comparison, we performed another PCA with the data set containing just terminal and ancestral ontogenetic trajectories and calculated the angles and lengths of the trajectories on the basis of the first two principal components, which were found to contain all significant shape information based on the broken stick method (see above).

Finally, the ancestral shape reconstructions calculated for the adult representatives of the taxa were used to qualitatively discuss the evolutionary changes within basal Sauropodomorpha and Theropoda with respect to the ontogenetic changes and heterochronic trends found in the different trajectories.

## RESULTS

### General ontogenetic changes

The juveniles of the sauropodomorph *Massospondylus* and the theropods that were sampled here tend to have skulls with a short and abruptly tapering snout, short antorbital fenestrae, large subcircular orbits, slender jugals, and dorsoventrally deep orbital and postorbital regions relative to the snout. In addition, the jaw joint is more anteriorly placed relative to the occiput, with exception of the juvenile specimen of *Allosaurus* sampled here. The general ontogenetic pattern includes an elongated and dorsoventrally deeper snout relative to the orbital and postorbital regions, and also a relative increase in size of the antorbital fenestra, which correlates with a relative decrease in size of the orbit. Finally, the jugal becomes more massive in all taxa, which is more pronounced in the large-bodied theropods *Allosaurus* and *Tarbosaurus* (Fig. 1). The relative elongation of the snout and antorbital fenestra were not observed in the *Allosaurus* or *Tarbosaurus* ontogenies, which is probably due to the fact that the juveniles sampled do not represent the earliest ontogenetic stages (*Loewen, 2009*; *Tsuihiji et al., 2011*, see 'Discussion'). However, the discovery of an isolated maxilla identified as a hatchling allosauroid might indicate that the snout of early *Allosaurus* juveniles was probably short and subsequently increased in relative length during early ontogeny (*Rauhut & Fechner, 2005*).

In addition to these more general ontogenetic modifications, individual taxa show specific shape changes (Fig. 1):

(a) In *Massospondylus* the external naris becomes larger and expands dorsally. The postorbital also becomes relatively more robust. The infratemporal fenestra decreases in relative size. The jaw joint moves anteroventrally.

(b) In *Coelophysis* the external naris becomes smaller and shifts anteriorly. The notch of the alveolar margin between the premaxilla and maxilla decreases in relative size during ontogeny, while the alveolar margin of the premaxilla becomes more aligned with that of the maxilla. The descending process of the lacrimal becomes more slender anteroposteriorly. The postorbital becomes more gracile in its relative shape. The infratemporal fenestra increases in relative size. The jaw joint moves posterodorsally.

(c) In the megalosaurid taxon, the external naris becomes relatively larger and expands posteriorly. The lacrimal is inclined strongly backwards and the postorbital becomes

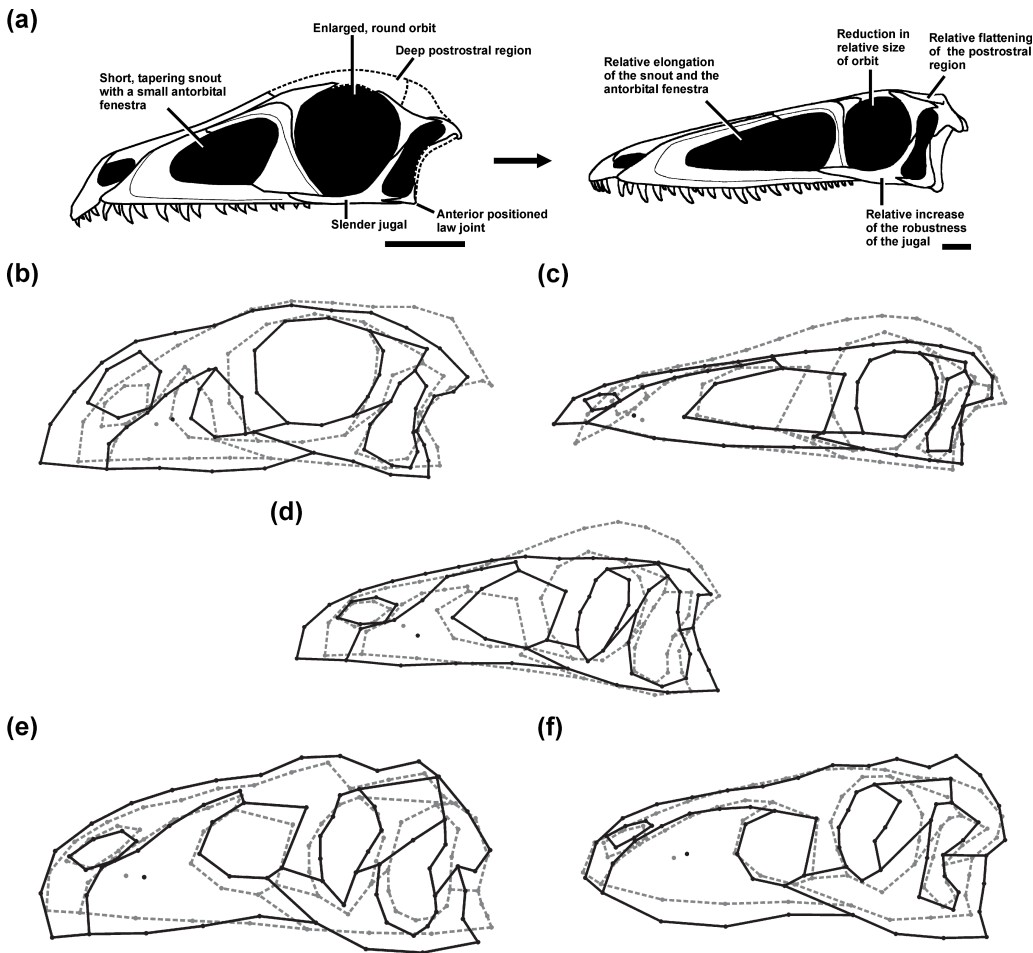

**Figure 1 Ontogenetic changes in the skull of saurischian dinosaurs.** (A) General ontogenetic pattern in Saurischia exemplified for the basal theropod *Coelophysis bauri* (adult specimen modified after *Rauhut, 2003*). (B–F) Specific ontogenetic changes in saurischian dinosaurs visualized as wireframes of Procrustes-fitted shapes. (B) *Massospondylus*. (C) *Coelophysis bauri*. (D) Megalosaurid taxon. (E) *Allosaurus*. (F) *Tarbosaurus*. Grey dashed lines represent the juvenile stage and black solid lines represent the adult stage.

relatively more robust. The infratemporal fenestra increases in its relative size. The jaw joint moves posteriorly.

(d) In *Allosaurus* the external naris does not change in relative size, but shifts ventrally. The descending process of the lacrimal becomes more massive anteroposteriorly. The lacrimal develops a prominent dorsal horn through ontogeny. In contrast to previous taxa, the postorbital region of *Allosaurus* increases dorsoventrally such that the postorbital, quadratojugal and squamosal become relatively more robust. The ventral shift of the jugal leads to the formation of a wide angle between the ventral margins of the maxilla and jugal. Due to its posteroventral expansion, the postorbital affects the shape of the infratemporal fenestra. However, the infratemporal fenestra does not decrease in its relative size, but shifts anteroventrally. The jaw joint moves anteroventrally.

(e) In *Tarbosaurus* the external naris does not change in relative size, but shifts dorsally. As in *Allosaurus*, the descending process of the lacrimal becomes more massive. The same is true for the postorbital region, which increases in depth dorsoventrally. This change is correlated with the development of a more robust postorbital, quadratojugal and squamosal. The jaw joint moves posteroventrally.

## Principal component analysis and phylogenetic correlation

The first three principal components account for 68.0% of the total variation (PC 1: 30.8%; PC 2: 23.9%; PC 3: 13.3%), in which PC 2 and PC 3 contain the main allometric shape information (see Table S12). PC 1 describes the overall skull depth, size and anteroposterior position of the external naris, length of the premaxilla, size of the maxillary antorbital fossa, and position of the lacrimal and postorbital on the anteroposterior axis (affecting the size of the antorbital fenestra, orbit and infratemporal fenestra). The dorsoventral dimension of the orbit is affected by the relative depth of the entire orbital and postorbital regions, while that of the infratemporal fenestra is affected by the relative position of the jugal-quadratojugal bar. The variation in the depth of the skull also affects the position of the jaw joint on the dorsoventral axis (Fig. 2C). PC 2 describes the length of the snout caused by variation in the length of the maxilla and inclination and anteroposterior position of the lacrimal. The inclination of the lacrimal affects the size of the antorbital fenestra, while both position and inclination affect the anteroposterior dimension of the orbit. PC 2 also accounts for the length and the dorsoventral position of the external naris and size of the upper temporal region (Fig. 2C). PC 3 describes the length of the premaxilla, posterior extension of the external naris, dorsoventral height of the maxilla, and anteroposterior dimension of the ventral process of the lacrimal (which affects the shape of the antorbital fenestra and orbit). The shape of the orbit is further affected by the anteroposterior dimension of the jugal-postorbital bar. Further variation captured by PC 3 is related to the shape of the skull roof in the orbital and postorbital regions, dorsoventral height of the infratemporal fenestra, and position of the jaw joint on the anterodorsal-posteroventral axis (Fig. 2C).

The permutation tests and the multivariate $K$ statistic recovered that both Procrustes-fitted shapes (tree length weighted by branch lengths = 0.5108, $p < 0.0001$; $K = 0.2607$, $p = 0.0016$) and centroid size (tree length weighted by branch lengths = 8.3598, $p = 0.0005$; $K = 0.8900$, $p = 0.0002$) are correlated with phylogeny. Furthermore, the multivariate regression analysis reveals that skull shape is significantly correlated with centroid size (correlation index: 15.32%, $p < 0.0001$) (Fig. 4A, Table S12).

## Ontogenetic trajectories in the PCA morphospace

Based on the PCA results of the original data set (i.e., including semi-landmarks), the ontogenetic trajectories are not uniform (Fig. 2 and Table 1). The trajectory of *Allosaurus* is short and mainly explained by shape variation captured by PC 1, while that of *Tarbosaurus* is also short, but mainly explained by PCs 1 and 3. The third principal component has stronger influence on the ontogenetic shape variation in *Tarbosaurus* based on the length of its trajectory. Compared to *Allosaurus* and *Tarbosaurus*, the other ontogenetic trajectories

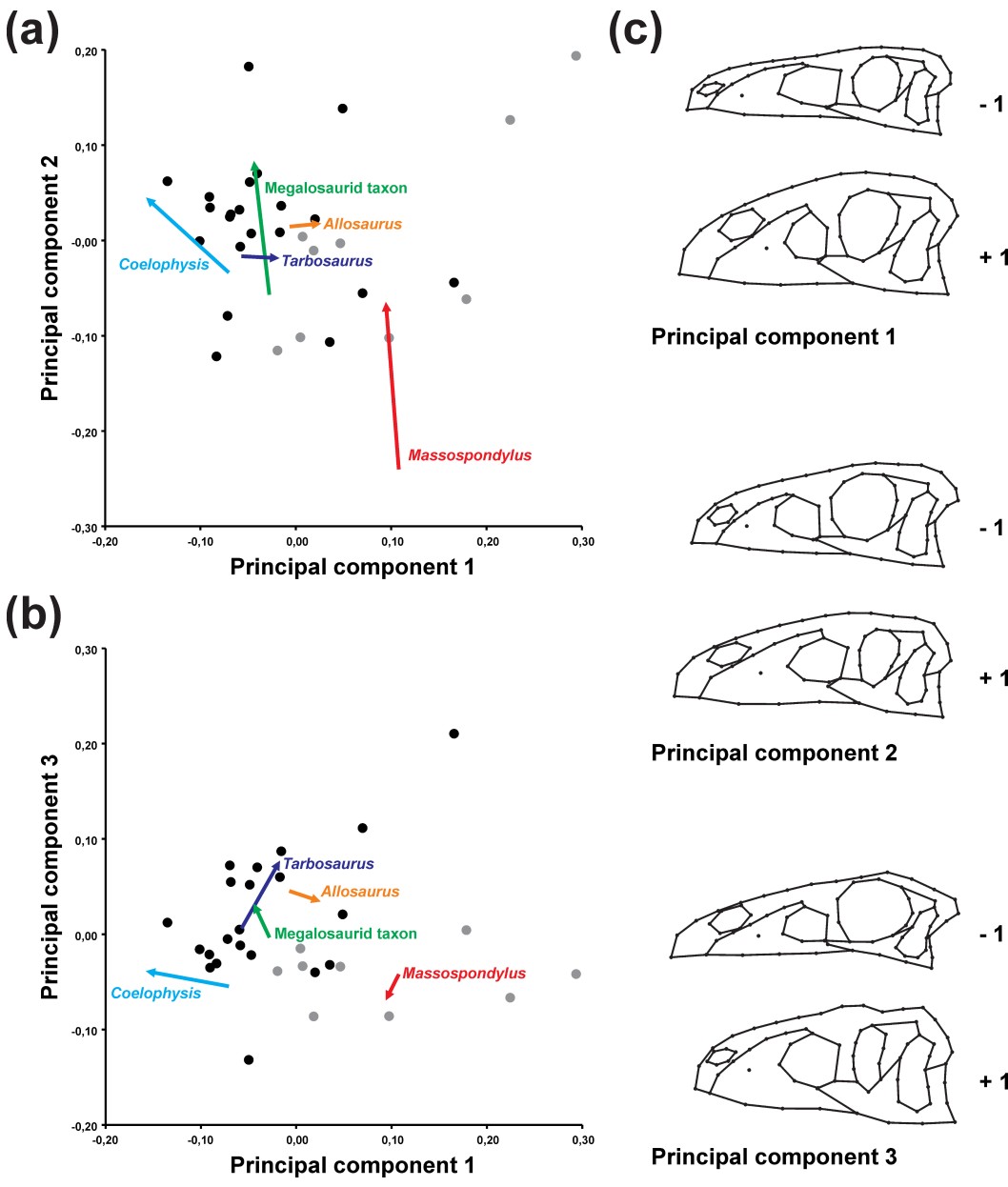

**Figure 2  Principal component analysis of the main sample.** (A) Ontogenetic trajectories of terminal taxa for PC 1 versus PC 2. (B) Ontogenetic trajectories of terminal taxa for PC 1 against PC 3. (C) Illustration of the main shape changes for the first three principal components. Theropod taxa are shown as black dots, while sauropodomorph taxa are shown as grey dots. The arrows illustrate the different ontogenetic trajectories, in which the arrowhead marks the position of the adult individual.

are longer. The trajectory of *Coelophysis* is mainly explained by the shape variation captured by PCs 1 and 2, while its slope is opposite to the direction along PC 1 compared to the trajectories of *Allosaurus* and *Tarbosaurus*. Based on the angles, the ontogenetic trajectories of *Massospondylus* and the megalosaurid taxon are mainly influenced by the shape variation captured by PCs 2 and 3, in which the ontogenetic trajectory of *Massospondylus* is directed

Table 1 **Angles and length of terminal ontogenetic trajectories.** Angles of ontogenetic trajectories against PC 1, pairwise angles between ontogenetic trajectories in the PC 1–PC 2 and PC 1–PC 3 morphospace and length of ontogenetic trajectories in the PC 1–PC 2 and PC 1–PC 3 morphospace (Figs. 2A and 2B). Green fields mark pairwise angles in the PC 1–PC 2 morphospace and orange fields mark that of the PC 1–PC 3 morphospace. Angles, lengths and slopes of ontogenetic trajectories versus log-transformed centroid size (LogCS) (Fig. 4A).

| | *Massospondylus* | *Coelophysis* | **Megalosaurid taxon** | *Allosaurus* | *Tarbosaurus* |
|---|---|---|---|---|---|
| Angle (PC 1–PC 2) | 85.6492 | 42.3458 | 83.3216 | 5.3228 | 3.7406 |
| Length (PC 1–PC 2) | 0.1761 | 0.1174 | 0.1414 | 0.0332 | 0.0403 |
| Angle (PC 1–PC 3) | 63.2316 | 10.1684 | 65.0464 | 18.5268 | 60.5157 |
| Length (PC 1–PC 3) | 0.0297 | 0.0881 | 0.0390 | 0.0349 | 0.0818 |
| *Massospondylus* | – | 73.4000 | 128.2780 | 98.2417 | 177.2841 |
| *Coelophysis* | 43.3033 | – | 54.8780 | 171.6416 | 109.3159 |
| Megalosaurid taxon | 2.3276 | 40.9757 | – | 133.4803 | 54.4379 |
| *Allosaurus* | 89.0280 | 132.3313 | 91.3556 | – | 79.0425 |
| *Tarbosaurus* | 98.0914 | 141.3947 | 100.4190 | 9.0634 | – |
| Angle (LogCS) | 3.3947 | 4.8961 | 4.6105 | 0.2535 | 1.5851 |
| Length (LogCS) | 2.2815 | 1.0636 | 1.8147 | 1.0657 | 1.4016 |
| Slope (LogCS) | 0.0593 | 0.0857 | 0.0806 | 0.0044 | 0.0277 |

in the opposite direction along PC 3 to that of the megalosaurid taxon and *Tarbosaurus.* However, the length of the trajectories indicates that the second principal component has major influence on the shape variation in both species during ontogeny.

The PCA reveals that the ontogenetic elongation of the snout is primarily related to a relative increase in the length of the maxilla (PCs 1, 2). In *Massospondylus* and the megalosaurid taxon the ontogenetic elongation of the snout is further affected by the relative increase of the length of the premaxilla (PC 3). The relative increase in snout depth results mainly from a ventral expansion of the maxilla, which is more prominent in *Allosaurus* and *Tarbosaurus* than in other taxa (PCs 1, 3). In the megalosaurid taxon and *Allosaurus,* maxillary deepening occurs together with a dorsoventral expansion of the nasal (PC 1). Additionally, dorsoventral expansion of the premaxilla is observed in *Allosaurus* and *Tarbosaurus* (PC 1). The relative elongation of the snout in *Massospondylus*, the megalosaurid taxon and *Coelophysis* correlates with a relative increase in the anteroposterior length of the antorbital fenestra, caused by a posterior shift of the lacrimal and elongation of the maxilla (PCs 1, 2). Additionally, in *Coelophysis* the anterior border of the antorbital fenestra extends anteriorly (PC 1). In both *Massospondylus* and the megalosaurid taxon, the antorbital fenestra is shifted posteriorly during ontogeny (PC 2). The megalosaurid taxon shows a further dorsal expansion of the antorbital fenestra (PC 3), not seen in the latter two taxa. Although no relative size changes could be observed in the antorbital fenestrae of *Allosaurus* and *Tarbosaurus*, the antorbital fenestra of *Allosaurus* shifts posterodorsally during ontogeny, whereas that of *Tarbosaurus* shifts ventrally. In most trajectories, the most anterior point of the antorbital fossa shifts posteriorly during ontogeny (PCs 1–3), but a relative decrease in the length of the maxillary antorbital fossa is present in *Allosaurus* and

*Tarbosaurus* (PC 1). In the megalosaurid taxon, the anterior margin of the antorbital fossa shifts ventrally, whereas in *Coelophysis* it shifts anteriorly (PC 1), which correlates with the anterior elongation of the antorbital fenestra in this taxon (see above). As mentioned above, the orbit decreases in relative size in all taxa during ontogeny (PCs 1–3). In *Coelophysis* and *Massospondylus* this is related to a relative shift of the lacrimal posteriorly (PCs 1, 2). In the megalosaurid taxon, *Allosaurus* and *Tarbosaurus* the relative size reduction is correlated with a change in orbital shape from subcircular to oval. In the megalosaurid taxon these changes are linked to a posterior shift of the lacrimal (PC 2) and anterior shift of the postorbital and ascending process of the jugal (PC 3), which is correlated with an anterior extension of the infratemporal fenestra. In *Allosaurus*, the ontogenetic changes of the orbit are related to the posterior extension of the lacrimal and anterior shift of the postorbital and ascending process of the jugal (PC 1). Additionally, the orbit of *Allosaurus* is shifted slightly dorsally. In *Tarbosaurus*, these changes result from an anterior extension of both the postorbital and ascending process of the jugal (PC 3). The orbit of *Tarbosaurus* becomes posteriorly constricted by an anterior shift of the ventral process of the postorbital, forming a suborbital process.

We examined the differences in the trajectory directions when terminal and ancestral ontogenetic series are compared to each other (Fig. 3 and Table 2). The significant shape variation evaluated via the broken stick method is described by the first two principal components (PC 1: 50.39%; PC 2: 20.79%). Both axes are correlated with centroid size (see Table S12). The ontogenetic trajectory of *Coelophysis* in mainly influenced by PC 1, while that of the megalosaurid taxon, *Massospondylus* and all ancestral trajectories is influenced by both PC 1 and 2, in which the first principal component is found to have a higher impact on the shape variation during ontogeny. In contrast, the ontogenetic trajectories of *Tarbosaurus* and *Allosaurus* are mainly influenced by PC 2.

## Ontogenetic trajectories in the regression analyses

The ontogenetic trajectory of *Massospondylus* is longer than that of the hypothetical ancestor of Saurischia for both shape variables (regression score and Euclidean distance), while the values of the shape variables are significantly lower. However, the slope of the trajectory of *Massospondylus* based on the regression score is less pronounced than that of the saurischian ancestor, while it is more pronounced for the Euclidean distance (Figs. 4B, 4C, Tables 3 and 4). In contrast, the ontogenetic trajectory of the hypothetical ancestor of Neotheropoda is slightly longer and has a greater slope, while the regression score and the Euclidean distance of the adult individual are significantly higher than that of the saurischian ancestor. *Coelophysis* possesses a longer and steeper ontogenetic trajectory for both shape variables with significantly higher values than the hypothetical ancestor of Neotheropoda (Figs. 4B, 4C, Tables 3 and 4). The ontogenetic trajectory of the hypothetical ancestor of Orionides is shorter and has a lower slope than that of the neotheropod ancestor. The regression score of the adult individual is significantly higher, while the Euclidean distance is lower, but not significantly different. Compared to the hypothetical ancestor of Orionides, the megalosaurid taxon has a longer and steeper ontogenetic trajectory, with a significantly higher value for both shape variables

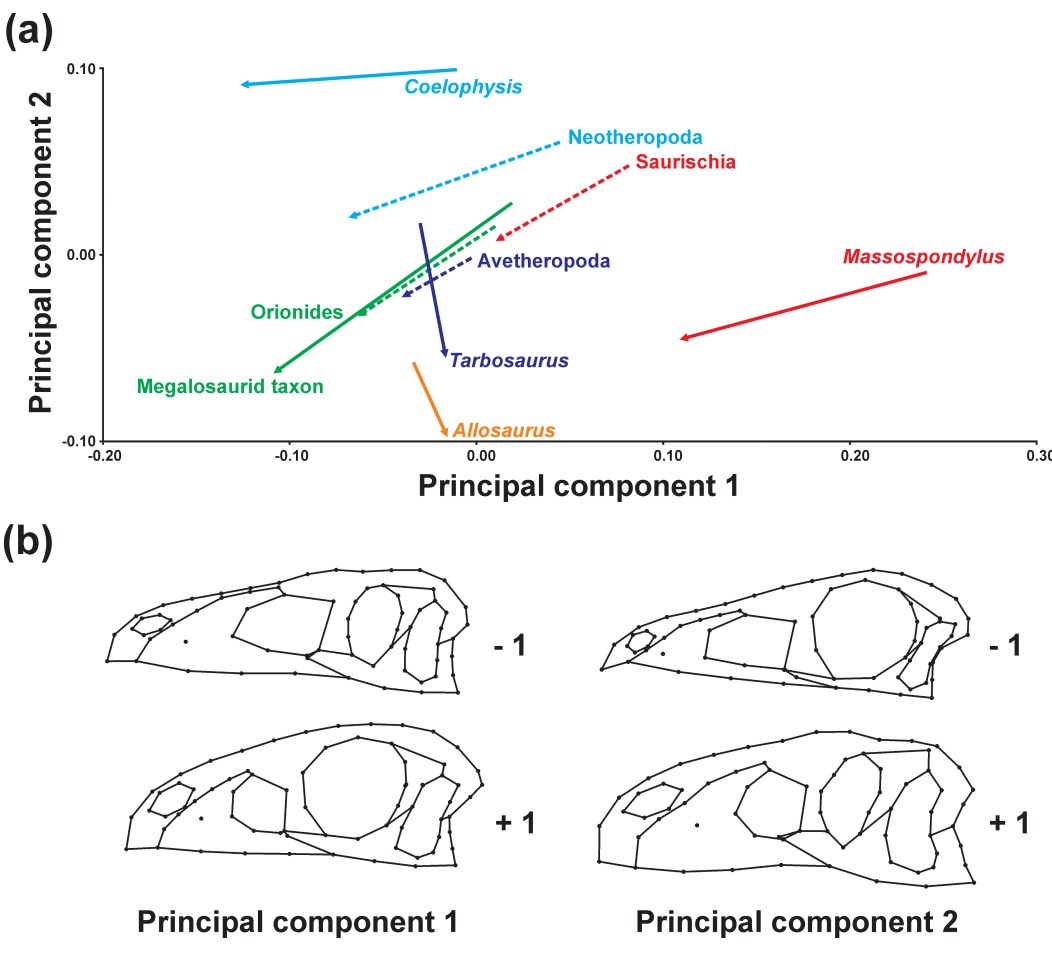

**Figure 3** **Principal component analysis of ontogenetic trajectories.** (A) Terminal and ancestral onto­genetic trajectories for PC 1 against PC 2. The arrows illustrate the different ontogenetic trajectories, in which the arrowhead marks the position of the adult individual and the base of the arrow indicates the ju­venile individual. (B) Illustration of the main shape changes for the first two principal components.

(Figs. 4B, 4C, Tables 3 and 4). In contrast, the ontogenetic trajectory of the hypothetical ancestor of Avetheropoda is shorter, possessing a lower slope and significantly lower regression score and Euclidean distance for the adult individual, when compared to the hypothetical ancestor of Orionides. The ontogenetic trajectories of *Allosaurus* and *Tarbosaurus* are longer than that of the hypothetical ancestor of Avetheropoda. Both trajectories show a slope decrease compared to their common ancestor. Interestingly, the slope is almost zero when the Euclidean distance is applied as shape variable, indicating only minor shape changes during the ontogeny as sampled. For *Allosaurus*, both shape values of the adult individual are higher than that of the ancestor, but only the regression score is significant. In contrast, the regression score of the adult individual of *Tarbosaurus* is significantly lower than that of the hypothetical ancestor of Avetheropoda, while the Euclidean distance results in a higher, but non statistically significant value (Figs. 4B, 4C, Tables 3 and 4).

**Table 2 Angles and lengths of terminal and ancestral ontogenetic trajectories.** Angles of ontogenetic trajectories against PC 1, pairwise angles between ontogenetic trajectories in the PC 1–PC 2 morphospace and length of ontogenetic trajectories in the PC 1–PC 2 morphospace (Fig. 3A).

| | Saurischia | Massospondylus | Neotheropoda | Coelophysis | Orionides | Megalosaurid taxon | Avetheropoda | Allosaurus | Tarbosaurus |
|---|---|---|---|---|---|---|---|---|---|
| Angle (PC 1–PC 2) | 29.5357 | 15.193 | 19.6691 | 4.0256 | 33.2773 | 35.5725 | 29.4664 | 65.5478 | 79.1993 |
| Length (PC 1–PC 2) | 0.082 | 0.1372 | 0.1202 | 0.1162 | 0.0879 | 0.1571 | 0.0429 | 0.044 | 0.0735 |
| Saurischia | – | | | | | | | | |
| Massospondylus | 14.3427 | – | | | | | | | |
| Neotheropoda | 9.8666 | 4.4761 | – | | | | | | |
| Coelophysis | 25.5101 | 11.1674 | 15.6435 | – | | | | | |
| Orionides | 3.7416 | 18.0843 | 13.6082 | 29.2517 | – | | | | |
| Megalosaurid taxon | 6.0368 | 20.3795 | 15.9034 | 31.5469 | 2.2952 | – | | | |
| Avetheropoda | 0.0693 | 14.2734 | 9.7973 | 25.4408 | 3.8109 | 6.1061 | – | | |
| Allosaurus | 84.9165 | 99.2591 | 94.783 | 110.4266 | 81.1749 | 78.8797 | 84.9858 | – | |
| Tarbosaurus | 71.265 | 85.6076 | 81.1315 | 96.7751 | 67.5234 | 65.2282 | 71.3343 | 13.6515 | – |

Foth et al. (2016), *PeerJ*, DOI 10.7717/peerj.1589

**Table 3 Angles and lengths of terminal and ancestral ontogenetic trajectories.** Angles, lengths and slopes of ontogenetic trajectories from the regression of shape (Regression score, RS and Euclidean Distance, ED) versus log-transformed centroid size (LogCS) (Figs. 4B and 4C).

| | Saurischia | *Massospondylus* | Neotheropoda | *Coelophysis* | Orionides | Megalosaurid taxon | Avetheropoda | *Allosaurus* | *Tarbosaurus* |
|---|---|---|---|---|---|---|---|---|---|
| **Regression (RS)** | | | | | | | | | |
| Angle (LogCS) | 4.3762 | 3.8814 | 5.1181 | 6.056 | 4.1743 | 5.029 | 3.0083 | 1.1845 | 0.7153 |
| Length (LogCS) | 1.1084 | 2.2828 | 1.3988 | 1.0657 | 1.3267 | 1.8158 | 0.9628 | 1.0659 | 1.4011 |
| Slope (LogCS) | 0.0765 | 0.0678 | 0.0896 | 0.1061 | 0.0730 | 0.0880 | 0.0526 | 0.0207 | 0.0125 |
| **Regression (ED)** | | | | | | | | | |
| Angle (LogCS) | 3.4145 | 5.0440 | 4.0199 | 5.0905 | 2.7768 | 3.4451 | 1.7014 | −0.1758 | −0.1087 |
| Length (LogCS) | 1.1071 | 2.2864 | 1.3967 | 1.0640 | 1.3248 | 1.8121 | 0.9619 | 1.0657 | 1.4010 |
| Slope (LogCS) | 0.0597 | 0.0883 | 0.0703 | 0.0891 | 0.0485 | 0.0602 | 0.0297 | −0.0031 | −0.0019 |

**Table 4 Overview of heterochronies in saurischian skull shape.** The differences of the regression scores (ΔRS) and the Euclidean distances (ΔED) between ancestor-descendent relationships of adult individuals from the regression analysis (Figs. 4B and 4C) and the interpretation regarding heterochrony.

|  | ΔRS | ΔED | Heterochrony |
|---|---|---|---|
| Saurischia-*Massospondylus* | −0.0262 | −0.0446 | Paedomorphosis |
| Saurischia-Neotheropoda | 0.0629 | 0.0733 | Peramorphosis |
| Neotheropoda-*Coelophysis* | 0.0140 | 0.0668 | Peramorphosis |
| Neotheropoda-Orionides | 0.0146 | (−0.0079) | NA |
| Orionides-megalosaurid taxon | 0.0507 | 0.0497 | Peramorphosis |
| Orionides-Avetheropoda | −0.0299 | −0.0256 | Paedomorphosis |
| Avetheropoda-*Allosaurus* | 0.0153 | (0.0066) | NA |
| Avetheropoda-*Tarbosaurus* | −0.0145 | (0.0015) | NA |
| 95% CIs | 0.0078 | 0.0098 | |
| Significance levels ($p = 0.05$) | 0.0117 | 0.0147 | |

**Notes.**

ΔRS and ΔED values in brackets mark insignificant trends.

NA, not available.

Based on the regression analysis, taxa with higher regressions scores tend to have elongated skulls with long and slender snouts that have a rounded anterior end, and possess anteroposteriorly long antorbital fenestrae, oval orbits and a post-rostrum only slightly dorsoventrally higher than the snout. The maxilla increases in its relative length, but also expands ventrally. The ascending process of the maxilla, the anterior and ascending processes of the jugal, and postorbital become more massive. In contrast, low regression scores account for skull shapes where these features are less pronounced, developed or even show opposite trends. When compared to the regression analyses containing all taxa, the relative position, length and slopes of the ontogenetic trajectories of the terminal taxa is almost identical (Fig. 4A), supporting the robustness of the results recovered.

## DISCUSSION

### Ontogenetic patterns

Our knowledge of the cranial ontogeny of non-avian dinosaurs remains fragmentary. Previous studies on cranial ontogeny have often been based on single species (*Gow, Kitching & Raath, 1990*; *Carr & Williamson, 2004*; *Horner & Goodwin, 2006*; *Hübner & Rauhut, 2010*; *Campione & Evans, 2011*; *Mallon et al., 2011*; *Canale et al., 2014*; *Frederickson & Tumarkin-Deratzian, 2014*), while only a small number of studies have investigated this topic on the interspecific level (*Carr, 1999*; *Evans, 2010*; *Bhullar et al., 2012*; *Mallon, Ryan & Campbell, 2015*). As is common in other animal groups, closely related species often undergo similar ontogenetic changes (see *Evans, 2010*; *Mallon, Ryan & Campbell, 2015*), while ontogenetic trajectories become more different with increased phylogenetic distance (see *Bhullar et al., 2012*) or in the case of a single taxon evolving extreme ontogenies compared to their relatives (*Horner & Goodwin, 2009*, see also *Erickson et al., 2004*). Despite the large phylogenetic distance between the ontogenetic series sampled here, the present study reveals that the cranial ontogeny of saurischian dinosaurs undergoes some general patterns, including the relative elongation and dorsoventrally heightening of the preorbital region, decrease in orbit size and increase in jugal robustness. However, the PCA shows that the different ontogenetic trajectories differ strongly in length, direction

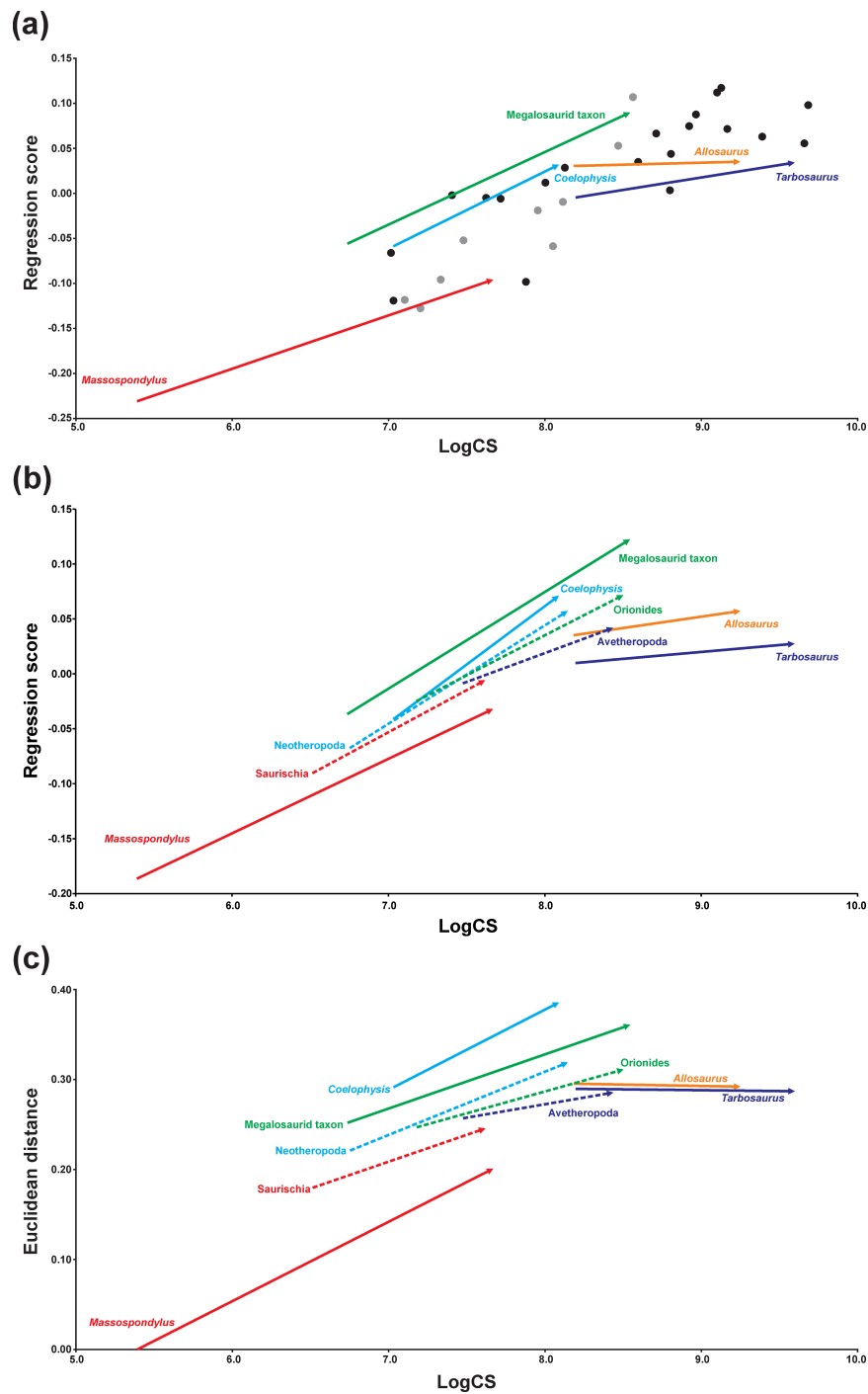

**Figure 4** **Centroid size regression analyses for the main sample.** (A) Regression analysis of all terminal taxa including ontogenetic trajectories against log-transformed skull centroid size (LogCS) ($p < 0.0001$). (B) Regression analysis of only terminal (solid arrows) and ancestral (dashed arrows) ontogenetic trajectories against log centroid size ($p < 0.0001$) using the regression score as shape variable. (C) Equivalent regression analysis to (B) using the Euclidean distance as shape variable. Theropod taxa are shown as black dots, while sauropodomorph taxa are shown as grey dots. The arrows illustrate the different ontogenetic trajectories, in which the arrowhead marks the position of the adult individual and the base of the arrow indicates the juvenile individual.

and also the location within the morphospace. Here, the theropod taxa are markedly separated from the sauropodomorph *Massospondylus*, which is morphologically very distinct from other basal sauropod taxa. This is especially obvious in the large distance within morphospace between *Massospondylus* and *Coelophysis*, which represent the most basal ontogenetic series of each clade indicating a strong diversification of skull shape in the early evolution of Saurischia. This may be related to differentiations along both lines of Saurischia in terms of ecology, including trophic specializations (see *Tykoski & Rowe, 2004*; *Barrett & Rayfield, 2006*; *Langer et al., 2010*; *Sakamoto, 2010*). The fact that the separation within the morphospace already take place among juvenile specimens indicates that these specializations might appear very early in ontogenetic development. Although the distances among such specimens in morphospace are large, the trajectories show that both species still share similar trends in cranial development (Figs. 2 and 3).

Although occupying a similar area of morphospace, the ontogenetic trajectory of the megalosaurid taxon differs markedly from that of *Allosaurus* and *Tarbosaurus*, showing more similarity with that of *Massospondylus* and *Coelophysis*, which share in common the relative elongation of the snout. The latter process probably represents a morphological trend within megalosaurids (*Therrien & Henderson, 2007*; *Sadleir, Barrett & Powell, 2008*), while large-bodied allosauroids and tyrannosaurids tend to have rather deeper than long skulls (see *Brusatte et al., 2012*; *Foth & Rauhut, 2013a*). However, as several medium-sized tyrannosauroids also have elongated snouts (*Li et al., 2010*; *Brusatte, Carr & Norell, 2012*; *Lü et al., 2014*; *Porfiri et al., 2014*), their ontogenetic trajectories would probably more closely resemble that of the megalosaurid taxon. One has to take into account that the length and direction of the ontogenetic trajectories of *Allosaurus* and *Tarbosaurus* are likely influenced by the fact that the juvenile specimens are ontogenetically more developed compared to that of the megalosaurid taxon (see below). Assuming that the hatchlings of *Allosaurus* and *Tarbosaurus* also had short, tapering snouts, the trajectory would probably be more similar in length and direction to that of the megalosaurid taxon.

## Heterochronic patterns

Previous workers have hypothesized that skull shape diversity in theropods and sauropodomorphs was driven by phylogenetic interrelationships, dietary preferences (*Young & Larvan, 2010*; *Brusatte et al., 2012*; *Foth & Rauhut, 2013a*), functional constraints (*Henderson, 2002*; *Foth & Rauhut, 2013a*), but also heterochrony (*Long & McNamara, 1997*; *Bhullar et al., 2012*). This study builds on the recent heterochronic analysis of *Bhullar et al. (2012)*, who primarily examined derived non-avian theropods and basal avians on the basis of a great number of ontogenetic trajectories of non-avian coelurosaurs and an extant phylogenetic bracket of crocodylians and birds, covering a broader scale of archosaurian craniofacial shape variation. However, by sampling and comparing ontogenetic trajectories of more basal saurischian taxa, our data set allows for reevaluation of the conclusions presented by *Bhullar et al. (2012)* with regards to basal sauropodomorphs, allosauroids and tyrannosauroids. The current study supports the influence of heterochrony on the cranial evolution of some saurischian lineages. When the differences of the regressions scores ($\Delta$RS) and the Euclidean distances ($\Delta$ED) in an ancestor-descendant relationship are compared,

the significant decrease of the shape values indicates potential paedomorphosis for the skull shape of *Massospondylus* and the hypothetical ancestor of Avetheropoda, while the skulls of *Coelophysis*, the megalosaurid taxon and the hypothetical ancestor of Neotheropoda, might be peramorphic. Thus, the current analyses support a paedomorphosis for basal sauropodomorphs as predicted by *Bhullar et al. (2012)*. Due to contradicting results regarding shape differences, no heterochronic pattern can be inferred for *Allosaurus*, *Tarbosaurus* and the hypothetical ancestor of Orionides. Thus, the current analyses do not support the predicted cranial peramorphosis for the allosauroids and tyrannosaurid lineage (*Long & McNamara, 1997*; *Bhullar et al., 2012*; *Canale et al., 2014*), while studies on growth (*Bybee, Lee & Lamm, 2006*; *Erickson et al., 2004*) and body size evolution (*Dececchi & Larsson, 2013*; *Benson et al., 2014*; *Lee et al., 2014*) in theropods indicate such a trend. However, this conflict is probably caused by incomplete sampling of ontogenetic trajectories, which affects the estimated shape of the hypothetical ancestor of Avetheropoda. A further expansion of the sampling of ontogenetic trajectories of saurischian taxa and the inclusion of an extant phylogenetic bracket (see *Bhullar et al., 2012*), would probably change some aspects of the analytical outcomes of this study (see below).

The increase in slopes in the ontogenetic trajectories of Neotheropoda, *Coelophysis* and the megalosaurid taxon, when compared to their ancestors, might show evidence for peramorphic acceleration. However, with a few exceptions, bone histology of basal theropods (e.g., *Coelophysis* and *Syntarsus*) is not well studied, so that this cannot be confirmed by growth patterns. Several studies on body size evolution support a peramorphic trend, showing an increase of size from the hypothetical ancestor of Saurischia over Neotheropoda towards megalosaurids (*Irmis, 2011*; *Dececchi & Larsson, 2013*). In contrast, the relative decrease in slope in the ontogenetic trajectory of the hypothetical ancestor of Avetheropoda might indicate neoteny. But again this cannot be confirmed by bone histological data at this time. *Dececchi & Larsson (2013)* and *Lee et al. (2014)* found a decrease of body size from the hypothetical ancestor of Tetanurae towards Avetheropoda, supporting a paedomorphic trend in body size. For *Massospondylus*, the situation is not entirely clear, as our two shape variables led to conflicting results regarding the slope, when compared with the saurischian ancestor. Thus, no underlying heterochronic process can be diagnosed for the paedomorphic skull shape of *Massospondylus*. Although basal sauropodomorphs show a gradual trend towards bigger body size (*Sander et al., 2010*; *Irmis, 2011*; *Benson et al., 2014*) and longer, accelerated growth (*Chinsamy, 1993*; *Erickson, Rogers & Yerby, 2001*; *Klein & Sander, 2007*), skull size decreased relatively (*Rauhut et al., 2011*). This relative shrinking might be the reason for the maintenance of a more juvenile skull shape in the early evolution of sauropodomorphs. However, due to the lack of information regarding the ontogenetic age of the individuals, the deduction of heterochronic process related to the slope (i.e., neoteny and acceleration) has to be considered with caution (see below).

The results of the regression analyses can be further used to interpret evolutionary shape changes found between hypothetical ancestors and terminal taxa in the ancestral shape reconstruction analyses of the main sample (i.e., continuous character mapping of the Procrustes-fitted shapes) in terms of paedomorphic or peramorphic trends (Fig. 5).

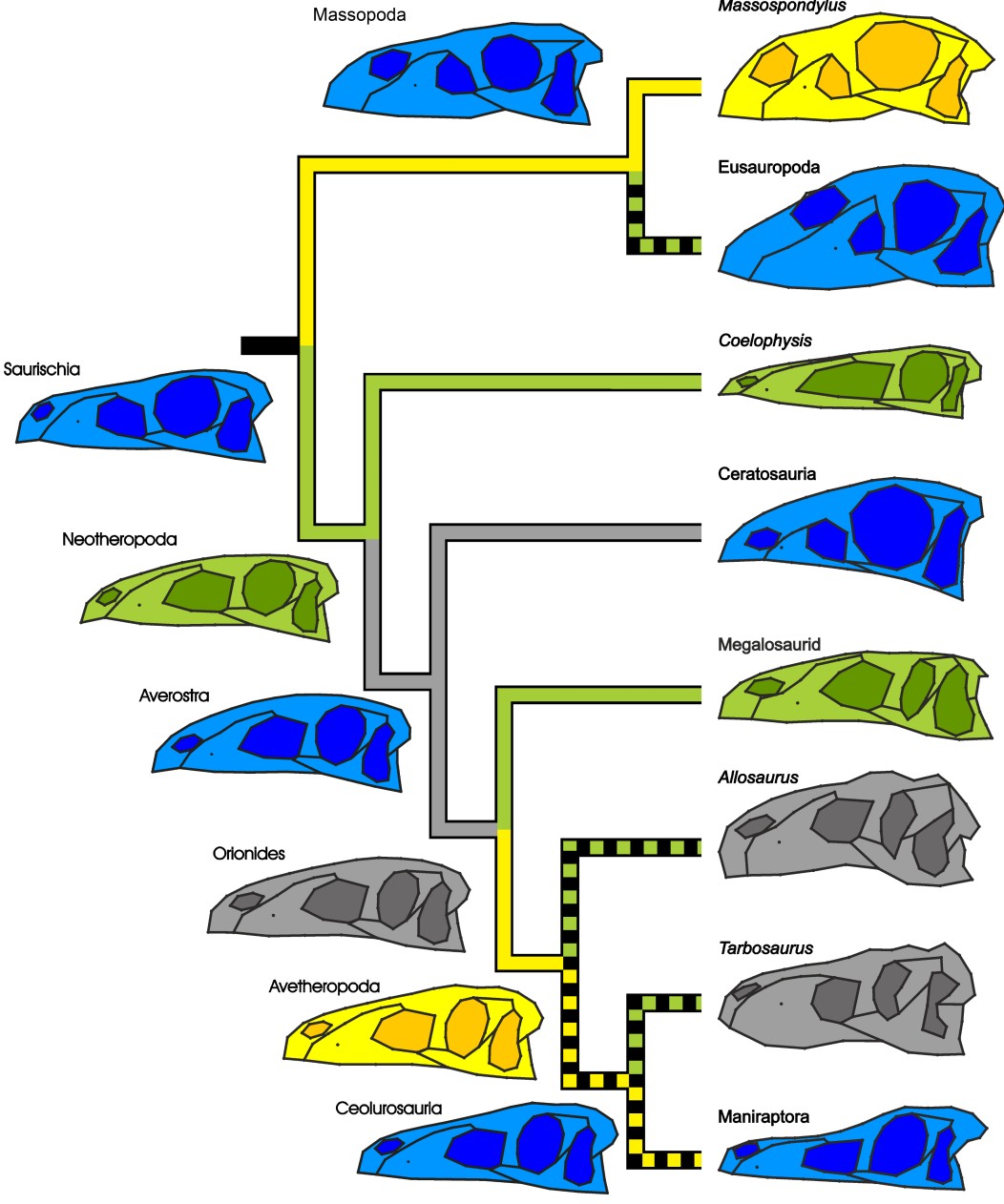

**Figure 5  Simplified phylogeny of Saurischia showing the main heterochronic trends of the skull.** Peramorphosis is colored in green and paedomorphosis in yellow. Grey trends indicate uncertain shape trends. Shape of the hypothetical ancestors based on the continuous character mapping of the Procrustes-fitted shapes of the adult terminal taxa from the original data set. Blue skulls represent ancestral skull shapes for which ontogeny could not be analysed. The heterochronic trends found in the regression analyses are visualized by the color of the branches. Possible heterochronic trends related to the skull evolution of allosauroids and basal coelurosaurs (see 'Discussion') are shown as dashed branches.

Comparing the skull shape of the hypothetical ancestor of Saurischia to that of Sauropodomorpha indicates a possible initial paedomorphosis in the evolution of the latter group as shown by the regression analyses, which is depicted by a decrease in the relative length of the preorbital region and an increase in the relative orbital size and depth of the postorbital region. As stated above, *Bhullar et al. (2012)* already proposed a cranial paedomorphosis for basal sauropodomorphs after finding a strong similarity between the skull shape of *Eoraptor* and the juvenile theropod *Coelophysis*, which had been also highlighted qualitatively by previous authors (e.g., *Ezcurra, 2007*). In addition, *Foth (2013)* has shown that the skull shape of *Eoraptor* and *Pampadromaeus* resembled that of the juvenile theropods *Sciurumimus* and *Juravenator*. In Eusauropoda the snouts become more aberrant due to a dorsal shift of the external naris, posterodorsal extension of the premaxilla, elongation of the ascending process of the maxilla and modification of the postorbital region, affecting the relative size of the jugal and postorbital, which become more gracile (*Wilson & Sereno, 1998*; *Rauhut et al., 2011*). While the shape changes in the snout and the shift of the naris were previously presumed to be peramorphic (*Long & McNamara, 1997*), one can assume on the basis of the current observations that the increase of gracility in the postorbital region of derived sauropods may result from modular paedomorphosis. In this context, *Salgado (1999)* has hypothesized that the reduction of the supratemporal fenestra and fusion of the frontals in diplodocoid sauropods is the result of a peramorphic heterochrony, while the loss of contact between squamosal and quadratojugal could be paedomorphic. However, these character changes are beyond the scope of the current study due to the lack of good skull material of juvenile individuals of basal sauropods, and thus, need to be analysed in more detail in future studies after the appropriate juvenile materials are discovered.

In contrast, the initial evolutionary changes in the skull shape of Theropoda were driven by peramorphic events, as is observed in *Coelophysis*, the megalosaurid taxon and the hypothetical ancestor of Neotheropoda in the regression analyses. These changes include the elongation of the snout, increase in length of the antorbital fenestra, and trends to a relatively smaller orbit and more robust post-rostral region. The basal ceratosaur *Limusaurus* has a rather small skull with a short snout, enlarged subcircular orbit and gracile jugal and postorbital, so it is possible that the more robust skull shape (oval orbit, massive jugal and postorbital) of large-bodied ceratosaurs like *Ceratosaurus* and abelisaurids (e.g., *Carnotaurus* and *Majungasaurus*) could be the result of a secondary peramorphosis as it was proposed for allosaurids and tyrannosaurids (e.g., *Long & McNamara, 1997*; *Bhullar et al., 2012*). However, due to the poor cranial knowledge and fluctuating phylogenetic relationships of basal ceratosaurs from the Early and Middle Jurassic (e.g., *Pol & Rauhut, 2012*; *Tortosa et al., 2013*), the early skull shape evolution of Ceratosauria is not currently reproducible. In contrast, the skull of the hypothetical ancestor of Avetheropoda is probably paedomorphic with respect to that of Orionides as shown in the regression analyses (Figs. 4B and 4C). This trend might extend to the hypothetical ancestor of Coelurosauria, Maniraptoriformes and Maniraptora, leading to a shorter, more tapering snout in lateral view, smaller antorbital fenestrae, enlarged subcircular orbits, and a more gracile postrostral region, resembling the skull shape of the juvenile megalosaurid *Sciurumimus*. These

findings may indicate that the paedomorphic trend hypothesized for Eumaniraptora by *Bhullar et al. (2012)* reaches back into the early evolution of Avetheropoda, and that basal coelurosaurs in fact represent "miniaturized" tetanurans, conserving juvenile characters in adult individuals. A similar trend is found for body size evolution in theropods, showing a successive decrease in body size within Avetheropoda (*Novas et al., 2012*; *Dececchi & Larsson, 2013*; *Lee et al., 2014*). In contrast to this supposed early paedomorphic trend, the ancestral shape reconstruction reveals that the skulls of allosauroids become secondarily more robust in relation to the hypothetical ancestor of Avetheropoda, supporting cranial peramorphosis (see *Canale et al., 2014*). This might also be the case for large-bodied tyrannosaurids (see *Long & McNamara, 1997*; *Bhullar et al., 2012*), although the current regression analyses could not find such a signal for both groups (see below). *Bhullar et al. (2012)* suggested a multi-step progenetic paedomorphosis for skull shape of Paraves and basal birds, with modular peramorphic trends related to beak formation, and further peramorphic trends for secondarily large-bodied troodontids and dromaeosaurids. These heterochronic changes were supported by trends regarding body size evolution (*Turner et al., 2007*; *Dececchi & Larsson, 2013*; *Lee et al., 2014*) and growth patterns (*Erickson et al., 2009*) found within Eumaniraptora. However, as it is the case for Sauropodomorpha, various trends seen in skull shape evolution of theropods need to be verified in the future regarding possible heterochrony on the basis of new material of both juvenile and adult specimens.

## Functional and ecological implications

The major differences in cranial shape found here clearly affect dietary preferences and functional constraints. The robust morphology of the postorbital region and the oval orbit in peramorphic skulls was previously discussed in relation to the generation of higher bite forces (*Henderson, 2002*; *Foth & Rauhut, 2013a*). However, these functional constraints go hand in hand with a decrease in cranial disparity (*Brusatte et al., 2012*). Paedomorphic changes in the orbital and postorbital regions were discussed in relation to visual elaboration and brain enlargement (*Bhullar et al., 2012*), and may have played an important role in nocturnal activity (*Schmitz & Motani, 2011*) or the evolution of flight within Paraves (*Balanoff et al., 2013*). On the other hand, large and circular orbits might simply correlate with reduced mechanical stresses during biting (*Henderson, 2002*), which have been suggested to also influence size and shape of the external naris, antorbital fenestra and infratemporal fenestra (*Witmer, 1997*; *Witzel & Preuschoft, 2005*; *Witzel et al., 2011*).

Both ontogenetic and phylogenetic variations in snout shape are likely related to dietary preferences (*Brusatte et al., 2012*; *Foth & Rauhut, 2013a*; *Foth, Bona & Desojo, 2015*; see above), in which the shape of premaxillae and maxillae partly determines the number and size of teeth (*Henderson & Weishampel, 2002*). Various examples of ontogenetic changes in the morphology and number of teeth are documented in Saurischia, including the basal sauropodomorph *Massospondylus*, coelophysoids (*Colbert, 1989*), basal tetanurans (*Rauhut & Fechner, 2005*; *Rauhut et al., 2012*), tyrannosaurids (*Carr, 1999*; *Tsuihiji et al., 2011*) and maniraptorans (*Kundrát et al., 2008*; *Bever & Norell, 2009*). Based on these observations the evolutionary increase in the number of teeth has been interpreted as peramorphic

(*Bever & Norell, 2009*). Tooth morphology, however, was found to be a stronger indicator of diet than the shape of the snout itself (see *Smith, 1993*; *Barrett, 2000*; *Barrett, Butler & Nesbitt, 2011*; *Zanno & Makovicky, 2011*; *Foth & Rauhut, 2013a*; *Hendrickx & Mateus, 2014*). In this context, *Rauhut et al. (2012)* hypothesised based on the similarities in the dentition of the juvenile megalosaurid *Sciurumimus*, adult compsognathids (*Stromer, 1934*; *Currie & Chen, 2001*; *Peyer, 2006*) and adult dromaeosaurids (*Xu & Wu, 2001*; *Norell et al., 2006*), that strongly recurved crowns with reduced or no mesial serrations may be paedomorphic in the latter two taxa. This heterochrony probably results from the decrease of body size observed in coelurosaurs (see above) and indicates an evolutionary shift in dietary preferences to smaller prey (see also *Zanno & Makovicky, 2011*).

## Limitations

As is common in vertebrate paleontology, the current study has a limited sample size when compared with extant neontological data sets (*Brown & Vavrek, 2015*). The current results are necessarily preliminary and must be viewed with caution especially because the sampling of ontogenetic trajectories is considerably lower than the sampling of adult individuals. Furthermore, trajectories are constructed using a single juvenile and adult specimen, with no intermediate forms. A single multistage example for *Tyrannosaurus* presented by *Bhullar et al. (2012)* has shown that during ontogeny the trajectory can change its direction considerably in a multivariate PCA plot. This, in turn, has an important impact on the length of the trajectory and its angle in relation to other trajectories. However, in regression analyses the difference with a two-stage approach should be less substantial as multivariate shape information is transformed into a single variable of shape for each stage with respect to its centroid size. The poor sample of juveniles is a result of rarity and poor preservation in the fossil record, which seems to be due to a number of factors, including preferred hunting of juveniles by predators (*Hone & Rauhut, 2010*) and a smaller likelihood of preservation, discovery, and collection because juveniles have smaller body sizes and more fragile bones than adults (*Brown et al., 2013*). Thus, due to small sample sizes, the statistical power of our analyses is generally low (see *Cumming, Fidler & Vaux, 2007*), limiting the explanatory power of our results. On the other hand, *Brown & Vavrek (2015)* recently demonstrated that the number of positive and negative allometries is underestimated in smaller samples in both paleontological and neontological data sets.

Another issue affecting our results is that the juvenile individuals sampled here are all of different early ontogenetic stages. The juvenile *Massospondylus* represents a composite of several embryos close to hatching (*Reisz et al., 2010*); the megalosaurid taxon (i.e., *Sciurumimus albersdoerferi*) is an early juvenile and its exact age could not be determined (*Rauhut et al., 2012*); the age of the *Coelophysis* juvenile reconstructed is approximately one year old (estimated by *Colbert, 1990*; *Rinehart et al., 2009*); the juvenile *Tarbosaurus* specimen is two to three years old (*Tsuihiji et al., 2011*); and the juvenile *Allosaurus* is likely five to seven years old (estimated based on *Bybee, Lee & Lamm, 2006*; *Loewen, 2009*). Thus, the different ontogenetic stages of the juvenile specimens and the small number of individuals for each ontogenetic series most likely affected the length, but maybe also the slope of the calculated trajectories (and thus the angles between the trajectories)

(see *Cardini & Elton, 2007*), including that of the hypothetical ancestors. Furthermore, the uncertainty regarding the age of the specimens leads to another weak point, as specimen age was not used to characterize the ontogenetic trajectories (see above), which is a common problem in paleontology (e.g., *McKinney, 1986*; *Klingenberg, 1998*; *Gould, 2000*; *Schoch, 2010*; *Bhullar et al., 2012*). In consequence, the applied regression analyses explored allometry and not heterochrony (see *Klingenberg & Spence, 1993*; *Klingenberg, 1998*). The substitution of age by size, however, would imply similar growth dynamics (i.e., proportionality between age and size) between ancestors and descendants, which would consequently ignore heterochronic processes related to growth rates (i.e., progenesis and acceleration). Although dinosaurs generally have higher growth rates compared to other non-avian reptiles, histological studies reveal that growth rates are not identical (*Erickson, Rogers & Yerby, 2001*; *Erickson et al., 2004*; *Padian, De Ricqlès & Horner, 2001*; *Sander et al., 2004*; *Erickson et al., 2009*; *Grady et al., 2014*; *Werner & Griebeler, 2014*). Therefore, allometric patterns cannot be used to infer heterochrony beyond paedomorphosis and peramorphosis as argued by *Klingenberg & Spence (1993)* and *Klingenberg (1998)*. Taking the uncertainties related to the lengths and slopes of the ontogenetic trajectories (due to incomplete ontogenetic series) and statistical uncertainties (due to the small sample size) into account, the classifications of underlying heterochronic processes would be misleading and probably erroneous.

In the current study, the interpretations of paedomorphosis and peramorphosis rely on the significant shape differences between adult individuals of the ontogenetic trajectories expressed by shape vectors in the regression analyses, for which the multivariate shape data were transformed into a univariate shape variable. These differences are affected by type of shape variable, but more importantly by the ancestral shapes, which in turn depend on the phylogenetic relationships, the algorithm of time calibration (e.g., *Bapst, 2014*) and the method of reconstruction (e.g., *Martins, 1999*; *Webster & Purvis, 2002*). Thus, one has to be aware that the application of different methods could result in slightly different ancestral shapes, affecting the value of the shape variable. However, because the current sample covers all major linages of basal saurischians except of crested taxa, which were found to impact the ancestral shape of the skull roofs significantly (see Fig. S5 and Table S6), the results of the ancestral reconstruction of adult individuals are viewed as valid. By using two different shape variables (Regression score and Euclidean distance), it was possible to confirm significant results through multiple methods.

The undefined trend found for *Tarbosaurus* in relation to the hypothetical ancestor of Avetheropoda illustrates the limitations of our analyses. Our result is seemingly contradictory to previous hypotheses and our ancestral shape reconstruction, which proposed peramorphosis as the main driver of skull evolution in large-bodied tyrannosaurids (see above, *Long & McNamara, 1997*; *Bhullar et al., 2012*). As stated above, this result is most likely related to the small sample size of ontogenetic trajectories as skulls with elongated and slender snouts are considered to be peramorphic on the basis of the regression analyses. The inclusion of more ontogenetic trajectories of large-bodied theropods would probably change this result in favour of a trend towards a deeper snout. Furthermore, large-bodied tyrannosaurids like *Tarbosaurus* descended

from small-bodied coelurosaurian ancestors (*Xu et al., 2004*; *Xu et al., 2006*; *Brusatte et al., 2010*; *Rauhut, Milner & Moore-Fay, 2010*; *Benson et al., 2014*), which means that the hypothetical inclusion of an ontogenetic trajectory of a small-bodied basal coelurosaur (e.g., *Compsognathus*, *Dilong*, *Haplocheirus*) and a respective hypothetical ancestor of Coelurosauria would probably change the current results, leading to a secondary peramorphic trend in Late Cretaceous tyrannosaurids, as suggested by previous authors. Thus, this result is very likely an artefact of incomplete sampling. In this context, the limited number of ontogenetic series of basal sauropodomorphs results only in a rough trend regarding the relationship between cranial ontogeny and evolution, which cannot be extended to more general patterns in the skull shape evolution of basal sauropods.

## CONCLUSIONS

The importance of heterochrony in non-avian dinosaur skull evolution is a relatively new concept (see *Long & McNamara, 1997*; *Bhullar et al., 2012*). This study quantitatively assesses the impact of skull heterochrony across early saurischian evolution, allowing testing some of the heterochronic trends proposed by *Bhullar et al. (2012)* and further highlights different vantages of using morphometric data to elucidate heterochronic trends. We estimated hypothetical ontogenetic trajectories in Saurischia, Neotheropoda, Orionides, and Avetheropoda using ontogenetic trajectories of *Massospondylus*, *Coelophysis*, a megalosaurid taxon, *Allosaurus* and *Tarbosaurus*. When compared using PCA, the ontogenetic trajectories of the terminal taxa show great variation in length and direction, but still follow some very general patterns, including a relatively elongated and dorsoventrally deeper preorbital region, decrease in orbit size and increase in jugal robustness. General peramorphic skulls include more elongate and slender snouts, elongate antorbital fenestrae, oval orbits, dorsoventrally shallower post-rostral regions, and more massive maxillae, jugals, and postorbitals. Paedomorphic skulls show the opposite features. The shape changes from the hypothetical ancestor of Saurischia to *Massospondylus* were paedomorphic, as previously suggested by *Bhullar et al. (2012)*. In contrast, skull evolution of basal theropod taxa was probably affected by peramorphic trends. However, Avetheropoda showed paedomorphic changes compared to Orionides. This might indicate that the paedomorphic trend found for Eumaniraptora (see *Bhullar et al., 2012*) may reach back into the early evolution of Avetheropoda. The hypothesized peramorphic evolution for skull shape of allosaurids and tyrannosaurids could not be supported by the current study, but this probably resulted from the small sample size of ontogenetic trajectories. Although our data showed minimal differences between our crested-taxa and non-crested taxa data sets and semi-landmark and no semi-landmark data sets, it is important to fully evaluate all possible sources of trends, especially when working with a small data set. As stated above, our study is hampered by the preservation of the fossil record (mainly the poor sample of complete juvenile specimens) and more finds will help to elucidate other evolutionary patterns related to heterochrony. With a larger number of taxa comprising juvenile and adult stages it will be possible to further test heterochronic hypotheses within Saurischia in more detail, and eliminate artefacts related to sample size. Future studies may also examine ontogenetic

histories of individual taxa that have reasonably complete ontogenetic samples, such as *Coelophysis*, to evaluate which factors (dietary preference, heterochrony, etc.) drive shape change in individual taxa. A larger number of studies using geometric morphometrics for individual taxa as well as a more complete sampling within Saurischia are necessary to more completely assess the importance of heterochronic processes in both sauropodomorph and theropod skull shape. In addition, it would be of value to explore modularity in saurischian skulls to project the investigation of heterochronic processes to particular skull regions. In sum, this study demonstrates that heterochrony played an important role in basal non-avian saurischian skull evolution building upon previous studies (*Bhullar et al., 2012*).

**Institutional Abbreviations**

| | |
|---|---|
| **BMMS** | Bürgermeister Müller Museum Solnhofen, Solnhofen, Germany |
| **CM** | Carnegie Museum of Natural History, Pittsburgh, USA |
| **GR** | Ruth Hall Museum, Ghost Ranch, USA |
| **IVPP** | Institute of Vertebrate Paleontology and Paleoanthropology, Beijing, China |
| **MCZ** | Museum of Comparative Zoology, Harvard University, USA |

## ACKNOWLEDGEMENTS

We thank Oliver Rauhut (Bayerische Staatssammlung für Paläontologie und Geologie, München), Miriam Zelditch (University of Michigan), Johannes Knebel (Ludwig Maximilians University, München), Stefan Richter (University of Rostock), Walter Joyce and Eduardo Ascarrunz (both University of Fribourg) for discussion, and Michel Laurin (Sorbonne Universités, Paris) for comments on an earlier version of the manuscript. We further thank Matthew Lamanna (Carnegie Museum of Natural History, Pittsburgh), Alex Downs (Ruth Hall Museum, Ghost Ranch), David Gillette (Museum of Northern Arizona, Flagstaff) and Xu Xing (Institute of Vertebrate Paleontology and Paleoanthropology, Beijing) for access to collections. This study benefitted especially from critical comments of Jesús Marugán-Lobón (Universidad Autónoma de Madrid) and three anonymous reviewers.

### Funding

CF is supported by a DFG grant to Oliver Rauhut (RA 1012/12-1) and a postdoctoral fellowship of the DAAD German Academic Exchange Service (No. 9154678), BPH is supported by a Benjamin Franklin Fellowship at the University of Pennsylvania, MDE is supported by a grant of the DFG Emmy Noether Programme to Richard J. Butler (BU 2587/3-1). The funders had no role in study design, data collection and analysis, decision to publish, or preparation of the manuscript.

### Grant Disclosures

The following grant information was disclosed by the authors:
DFG: RA 1012/12-1.
DAAD German Academic Exchange Service: 9154678.
Benjamin Franklin Fellowship.
DFG Emmy Noether Programme: BU2587/3-1.

### Competing Interests

The authors declare there are no competing interests.

### Author Contributions

- Christian Foth conceived and designed the experiments, performed the experiments, analyzed the data, contributed reagents/materials/analysis tools, wrote the paper, prepared figures and/or tables, reviewed drafts of the paper.
- Brandon P. Hedrick and Martin D. Ezcurra conceived and designed the experiments, contributed reagents/materials/analysis tools, wrote the paper, reviewed drafts of the paper.

### Data Availability

All raw data are submitted as Supplemental Information, including a MorphoJ and Nexus file.

### Supplemental Information

Supplemental information for this article can be found online at http://dx.doi.org/10.7717/peerj.1589#supplemental-information.

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
