# Peer review of "Cranial ontogenetic variation in early saurischians and the role of heterochrony in the diversification of predatory dinosaurs"

_PeerJ, doi:10.7717/peerj.1589_

## Round 0.1 · original submission · Major Revisions

· Academic Editor

Major Revisions

The manuscript is a compelling quantitative evaluation of dinosaur skull variation at the interface between evolution (phylogeny) and development (ontogeny), interpreting the pattern of skull shape differences among dinosaur species under the scope of heterochrony. Reviewer 1 suggests that this study is very much like that of Bhullar et al., (2012) and I completely agree; the only real difference is the studied sample. As such, even if the subject is on dinosaur macroevolution, it is definitively clear that its results shall be of great importance for most evolutionary biologists, whether paleontologist or not, making it perfectly suitable for PeerJ.

I concur with the suggestions made by the reviewers, without exception, and I would strongly recommend that the authors consider this when revising their manuscript. I would stress paying special attention to the application of the technical issues raised by reviewer 2. The consideration that the analysis of single skull-portions or separate bone analyses might be unnecessary is particularly important, not only because it deviates the attention from the main findings, but also because it might involve methodological shortcomings. For instance, the fact that Procrustes superimposition is landmark-dependent makes it very unlikely that separate superimpositions of single bones (or delineated skull portions) yield equivalent results than those obtained by superimposing entire skull configurations.

Finally, although the study of heterochrony is important for dinosaur paleobiology per se, I’d personally stress that it is also very much so for the biological sciences as a whole, as it is an empirical proof for the connection between development and evolution (see e.g., Raff, 1996; The shape of Life). This should be stressed in the study, as though it may seem obvious for some biologists, it is certainly not for most scientists or even lay-people that have access to PeerJ (e.g., journalists, etc.). Further, I would also suggest that the authors consider simplifying their interpretations, not only for making their study more accessible, both to the specialist and broader readerships, but also because it is paramount to avoid the historical terminological, conceptual and practical confusion which has haunted this important topic for decades (please see, Gould 2000; Of Coiled oysters and big brains, in Evolution and Development). Namely, much of the work on heterochrony, especially in the 90’s, was hard to follow because definitions and methodologies weren't congruent among studies, thereby yielding misleading thoughts which definitively confused future avenues of research on the topic (see, Klingenberg, 1998, Heterochrony and allometry; Biol. Reviews). To avoid going back to such disengagement, it is essential that the authors make an stronger effort in specifying how their interpretations and terminology on heterchrony make sense through the lens of the methodology (Geometric Morphometrics), which indeed is rather new in the field, and definitively different from any other methodology (e.g., univariate) used to address heterochrony in the past (please see, Mitteroecker et al., 2005; Heterochrony and GM, Evolution and Development). Considering Bhullar et al., (2012) as a point of departure might help, given the similitude between approaches, though bearing in mind that interpreting more than one dimension on PCA morphospaces appears to be an essential part of this work. .

Reviewer 1 ·

Basic reporting

No Comments

Experimental design

No Comments

Validity of the findings

No Comments

Additional comments

The Foth et al. study examines the role of heterochrony in the early evolution of the saurischian dinosaur skull. The methodological approach is modern and theoretically sound, and the results/conclusions are well-reasoned. The paper is well written and the figures adequately convey the primary points of the study. My only substantive criticism is that the manuscript leaves me asking the question of how the results of this study differ from those of Bhullar et al. (2012). It is the certainly the case that the two studies emphasize different parts of the tree—with Bhullar et al. concentrating on the origin of the modern bird skull and the Foth study focusing on the early history of Saurischia. Foth et al. claim, however, that their study is the first to examine heterochrony in the early history of the saurischian skull, which is not exactly true. Bhullar et al included early saurischians and even the stem archosaur Euparkeria, so the hypotheses provided by that study encompass the part of the tree discussed by Foth et al. This point needs to be explicitly acknowledged and then the differences (including differences in taxonomic and methodological approach, as well as in resulting morphological changes and inferred processes) explicitly addressed. For example, Bhullar et al. argue that the early history of cranial evolution in theropods is dominated by changes that produced a more bird-like skull (not surprisingly, these changes include many of the same transformations discussed in the Foth et al. study). This evolutionary trend continued until roughly the origin of coelurosaurs (tyrannosaurid-grade theropods) where different pulses of paedomorphosis (and localized peramorphosis) then drove cranial evolution until the origin of the avian crown croup (and beyond). Does the overall pattern of paedomorphosis (Sauropodomorpha) and peramorphosis (early Theropoda) described in the Foth et al. study contradict the conclusions of Bhullar et al for the part of the tree shared between the two studies? If so, why would that be? Is it the addition of Massospondylus (fully bracketing the saurischian node), choice of landmarks, statistical approach, something else, or all of the above? Explicitly recognizing that the Bhullar et al. study does provide hypotheses for the early history of theropods , and that the Foth et al study is providing the first test of those hypothesis, places Foth et al. in a more accurate context that will only clarify the importance of the study to a generalized readership (and it shouldn’t take too much extra work).

A couple of relatively minor (but important) points.

Page 6, line 68. Ontogenetic series of vertebrate fossils actually aren’t that rare. Any extinct species represented by more than one specimen is likely to represent an ontogenetic series—albeit a series that generally encompasses a relatively small amount of that species’ ontogenetic change. This is not an unimportant point as it recognizes that almost every species can provide some information toward the larger question of how developmental timing effects evolutionary morphology.

Page 12, line 223. Should also cite Alberch et al., Fink 1982, Bhullar (2012), and perhaps even Balanoff and Rowe (2007) which used a tree-based approach to the study of skeletal development of an extinct theropod (Aepyornis).

Reviewer 2 ·

Basic reporting

This submissio nadhere to PeerJ policy. It is written in good english, it provides sufficeint background, the structure is well defined, figures are good but can be improved all raw data are provided although the authors should also upload a nexus file of the phylogeny with branch lenghts.

Experimental design

This is original, rigorous and of high technical standard.

Validity of the findings

The finding are robust althoguh because there are too many, it is hard to focus on the right ones. I have suggested some changes that might allow atuhors to focus more into the important details. All is reccomedned in more detailed in the general comments to the authors.

Additional comments

This is a welcome contribution on a very important topic in evolutioanry biology. The authors provides a series of data analyses to identify heterochronic changes in skulls of non-avian saurschian. The sample size is as adequate as could be for this kind of fossils that are rare. For this reason I cannot see any way for the authors to improve their sample that ideally should include multiple juveniles and adult stages for the species analysed. What I find hard to follow, is the presentation of the results and some graphics where it is unclear if points in morphospace represents juveniles, adults or single species averages. The authors should also focus less in their writing and describing anatomical changes of PC scores esepcailly if they show this only in the supplementary material. Also, the ontogenetic trajectories comaprison should be more explicit in the results sections as well as the tables about angle comparison. I reccomened the authors also to look a tthe form space becuase that might help them simplifying the protocol. As it is, I might reccomend some re-structuring of data analyses and results presentation:
1. Show in a morphospace juveniles vs adults only for the 5 taxa available
2. show in another morphospace size + shape of juveniles + adults of all species
3. show in a phylomorphospace adults only.
For phylogenetic signal I reccomend the use of K multiv now avaialble using the package R geomorph. This is not essential but will be more than welcome.
The regression of figures 4, 5 and 6 are all confusing and I think that the analyses of different modules is not needed unless the authorsd does not want to test modulariy in the adult sample only using RV Escoufier index...but that is definitely a topic for another paper. Figure 8 looks fine although it is unclear what the mixed colour inside the branches means?
Since these are 2D data, it might be also worth to see whattpsRelw outputs will be once you decide to allow the semilandmakrs to slide only. It might be time consuming but worth exploring.
Below I provide some minor points to condier:
Abstract
Line 36: “in light of heterochrony”…this is unclear. What do you mean? Also change next sentence into “Our results indicate that…”
Line 42: coelurosaurs may BE extended back into…”
Introduction
Line 58: You forgot an essential reference here: Gould, S. J. (1977). Ontogeny and phylogeny. Harvard University Press. You should discuss more in details how and why this is important. Consider also the work of “Mitteroecker, P., Gunz, P., & Bookstein, F. L. (2005). Heterochrony and geometric morphometrics: a comparison of cranial growth in Pan paniscus versus Pan troglodytes. Evolution & development, 7(3), 244-258” also for their methodological approach.
Line 81-82: saying that a topic is popular does not sound good…you can say that there has been an increasing interest in exploring skull shape of these animals using geometric morphmoemtircs…you should also summarise at least briefly the findings of some of these studes in line 85-88
Line 91: Another important reference now: “Adams, D. C., Rohlf, F. J., & Slice, D. E. (2013). A field comes of age: geometric morphometrics in the 21st century. Hystrix, the Italian journal of Mammalogy, 24(1), 7-14.”
Materials & Methods
Line 150-153: can you ensure us that your reconstruction is reliable? You can merge different areas using photographs and tpsSuper just to combine pieces of bone from different specimens. tpsSuper will merge the specimens and deform the pictures using GPA. There is also taphonomic deformation that should be corrected in some way.
Line 180: remove “were” before superimposed
Line 181: resizes “landmark coordinates of all the specimens….”
Line 190: the generated Procrustes “coordinates”
The methods described are good although in some instance too many approaches are described. The Adams and Collyer approach generally works well with two stages, while ontogeny is not exactly like that. You can eventually use the approach proposed by Adams and Collyer also for multistate but you should report differences in vector size, direction and shape. This is now implemented in the R package geomorph and you can try that. An example of this application is:
Meloro, C., Cáceres, N., Carotenuto, F., Sponchiado, J., Melo, G. L., Passaro, F., & Raia, P. (2014). In and out the Amazonia: evolutionary ecomorphology in howler and capuchin monkeys. Evolutionary Biology, 41(1), 38-51.
But again the gradient there is more ecologically rather than ontogenetic. For this reason it might be worth to try to follow the approach of Cardini, A., & Polly, P. D. (2013). Larger mammals have longer faces because of size-related constraints on skull form. Nature communications, 4. Essentially, you can use the form space by performing a PCA on procustes plus log Centroid Size. Then, looking after comparing angle vectors of different species-taxonomic groups within the same form space. It is not clear also based on your sample size how many adults and how many juveniles you have. Do you have for every species one adult and one juvenile? This needs to be more explicit in a table. By looking at your data carefully I realise that you essentially have juvenile and adult only for few taxa (5). This needs to be more explicit and clear in the text and in the figures
Line 226: what is an “informal” supertree? Do you mean you manually solved the topology based on literature? What if you use the Lloyd et al. (2008) dinosaur supertree?
Line 249: the morphoj phylo signal is interesting but it does not allow to properly understand how big is it unless you do not compute the homoplasy index. For this reason, I suggest you R, geomorph and the calculation of the K statistics separated for size and shape. Use in this case all adults 20 taxa.
Line 305-320: this is hard to read and imagine if we do not see the wireframe on your PCA graphs…also in your graph it is hard to understand where is the juvenile, where is the adult and where are the different species…so import this in excel and make your symbols more explicit or change it and make it bigger because as it is, I find this hard to follow.
Line 330-335: here the K multiv can help to understand if size or shape got the stronger signal…as it is, I find hard to compare 0.51 vs 8.35 tree length.
Line 371-390: this is too long and too detailed for things you are not showing Line 348: I think here it is more meaningful for you to show the angle comparison in a table
Discussion: this is fine but it will be good if you can break it in structure that relates more to the way you present the results. So first why differences (or similarity) in ontogenetic trajectories between the five species analysed.

---

## Round 0.2 · Major Revisions

· Academic Editor

Major Revisions

This new version of the manuscript by Foth and colleagues contains information that can be an interesting contribution of skull shape evolution and ontogeny in nonavian dinosaurs, but that possibly it is limited to that matter, without attemping to reach further.

While the methods have been correctly applied, the interpretations of the results remain straitjacketed to be heterochrony. I pointed this out in the previous version of the manuscript, and a new reviewer (an expert in in dinosaur paleobiology and Evo.-Devo) has said it even more clearly — if all is heterochrony, then perhaps nothing is heterochrony. I understand that taking into consideration the comments of both reviewers may be challenging, but I am sure that it will definitively become a useful contribution of geometric morphometrics in dinosaur paleobiology.

Reviewer 1 ·

Basic reporting

no comment

Experimental design

no comment

Validity of the findings

no comment

Additional comments

I appreciate the changes made to this version of the paper, including being more explicit about the relative lack of novelty in the approach and that these efforts are building on the Bhullar et al. paper. I also think that the changes made in response to the other reviewer's comments significantly improved the clarity of the results.

However, I have to admit that I am still confused as to how these results should be interpreted in light of those of Bhullar et al. The authors state that the Bhullar et al paper focuses on the crown-ward part of the theropod tree and that their analysis is vague with regards to early theropods. This seems to be somewhat misleading. My reading of the Bhullar et al study concludes that, in addition to examining taxa well outside of Dinosauria and within Maniraptora, that study also examined Coelophysis, Allosaurus, and Tarbosaurus. It is true that they don't appear to include Massospondylus or the same megalosaurid taxa, so there are taxonomic differences between the studies that might allow novel heterochronic changes to be identified in the early history of Saurichia. But, it seems that Bhullar et al. focus their discussion on the crown-ward, more bird-like, part of the tree, because that is where they recovered heterochronic events. This begs the question of why did that study not find the patterns that this study did for early theropods? Is it simply because
of the small difference in taxonomic sampling between the two studies--does including Massospondylus make that big a difference? Or, is there a significant
methodological difference between the two studies. The answers to these questions remain blaring unanswered in the Foth et al. manuscript and it seems to me that
unless these points are addressed, this study does more to confuse the evolution of ontogenetic trajectories in saurischians than it does to clarify them. I don't
mean this to sound harsh, I just think you are missing a great chance to increase the importance of these considerable efforts by actually building on the Bhullar et al. study, rather than acknowledging and then kind of dismissing it under the pretense that it is directed at the origin of birds. Almost all of the general morphological trends described in this study were also described in the Bhullar et al study--granted, in the supplementary information but that doesn't make them any less published. This poiint just highlights the importance of drawing a real distinction between what is original in this study and what is a duplication of the Bhullar study. Duplication is important for science but it needs to be explicitly recognized as such.

I was also wondering why you didn't include any ontogenetic series of non-saurischian taxa (even an croc)? Massospondylus is explicitly described as the outgroup, and I understand that the Saurischian ancestor can be reconstructed based only on its two descendant lineages, but wouldn't then interpreting the disparity between that ancestor and Massospondylus be relatively meaningless? How would you know if the Massospondylus trajectory is plesiomorphic for Saurischia? I may be missing something but it seems to me that if you want to say something meaningful about the saurischian node, you need an outgroup that is actually outside of that clade. If I am correct, then the deepest node you can actually discuss with your data is Neotheropoda.

Reviewer 3 ·

Basic reporting

This work has been reviewed several times already and I have little to add. There are no problems here.

Experimental design

Most of this work is a good contribution to the knowledge of shape evolution and ontogeny in nonavian dinosaurs. However, the inference of heterochrony is fundamentally flawed for several reasons. Chief among these is that comparing "regression score" is not sufficient for inferring peramorphosis or paedomorphosis. Slopes differ and it is clear that, if extended back to the perinatal stage, some of the trajectories would cross. What part then to actually compare? For better or for worse, the terms ending with "morphosis" refer to adult individuals. They are meant to refer to the y-axis (shape) position of the right-hand or "arrow" terminals of the authors' trajectories, no more and no less. This of course also assumes the y-axis is a good proxy for ontogenetic change, but that is a different discussion. These scenarios are extensively discussed by Alberch, Gould, Wake, Oster, and others.

So one is left with comparing the adult terminals, which, with the limited sample size available for nonavian dinosaurs, are going to show considerable jitter or noise. Moreover, the centroid size approximations may be very inaccurate because of the fossil record's bias toward larger taxa and because for certain clades in pivotal phylogenetic positions, such as megalosaurs and ceratosaurs, only large-bodied taxa are known. Note that if comparing terminal taxa, or trajectories, _any_ difference in shape at equivalent size will be classified as some kind of heterochrony. It does not seem reasonable to classify every single anatomical change as heterochrony -- this diminishes the power of the concept. In previous work on paedomorphosis leading to birds, there were some methodological problems. However, the major instance of heterochrony was supported by evidence from several sources, including histology and size change. In that work, unlike here, it could be shown that some early avialans had very similar trajectories to those of other archosaurs but that these trajectories were truncated. This suggested an actual mechanism as per Alberch et al. -- that of progenesis. This was the main suggestion of the paper. Note that the equivalent graph of trajectories could also be over-interpreted to assign some sort of heterochrony (at least according to the position of the adult endpoint) to every transition; as, I noted, could be done with any geometry of any group of organisms showing ontogenetic shape change. But if everything is heterochrony, then perhaps nothing is heterochrony. At the least, the overall trajectory-wide regression score criterion must be done away with, slope must be considered, and a much more nuanced view of conceptual work on heterochrony from the 1970s and 1980s must be taken, as well as a more realistic sense of the limitations of the data.

Validity of the findings

As I noted, there is much good material in here, but I do not think that heterochrony can be inferred the way the authors are attempting to do. A more sophisticated and nuanced interpretation of the limits of the data and of the implications of Alberch et al.'s work is required.

---

## Round 0.3 · Minor Revisions

· Academic Editor

Minor Revisions

Please consider revising the text to make it more accessible to a broader readership (see my attached PDF with some remaining suggested minor revisions).

---

## Round 0.4 · accepted · Accept

· Academic Editor

Accept

Congratulations to all the authors!